# Systems approach reveals photosensitivity and PER2 level as determinants of clock-modulator efficacy

Dae Wook Kim[1] , Cheng Chang[2,*] , Xian Chen[3], Angela C Doran[4], Francois Gaudreault[5], Travis Wager[6], George J DeMarco[7] & Jae Kyoung Kim[1,**]

## Abstract

In mammals, the master circadian clock synchronizes daily rhythms of physiology and behavior with the day–night cycle. Failure of synchrony, which increases the risk for numerous chronic diseases, can be treated by phase adjustment of the circadian clock pharmacologically, for example, with melatonin, or a CK1δ/ε inhibitor. Here, using *in silico* experiments with a systems pharmacology model describing molecular interactions, and pharmacokinetic and behavioral experiments in cynomolgus monkeys, we find that the circadian phase delay caused by CK1δ/ε inhibition is more strongly attenuated by light in diurnal monkeys and humans than in nocturnal mice, which are common preclinical models. Furthermore, the effect of CK1δ/ε inhibition strongly depends on endogenous PER2 protein levels, which differs depending on both the molecular cause of the circadian disruption and the patient's lighting environment. To circumvent such large interindividual variations, we developed an adaptive chronotherapeutics to identify precise dosing regimens that could restore normal circadian phase under different conditions. Our results reveal the importance of photosensitivity in the clinical efficacy of clock-modulating drugs, and enable precision medicine for circadian disruption.

**Keywords** circadian rhythms; CK1δ/ε inhibitor; personalized chronotherapy; systems pharmacology model

**Subject Categories** Computational Biology; Quantitative Biology & Dynamical Systems

**Mol Syst Biol. (2019) 15: e8838**

## Introduction

Circadian (~24 h) rhythms of diverse behavioral and physiological processes such as sleep and hormone secretion are coordinated by an endogenous timer, the circadian clock (Dibner *et al*, 2010). The key oscillatory molecular mechanism of the mammalian circadian clock centers around the PER1/2 proteins that mediate an autoregulatory negative feedback loop of their own transcription (Ukai & Ueda, 2010; Aryal *et al*, 2017; Takahashi, 2017). This generates circadian expression of PER1/2 at both the mRNA and protein levels. The period and phase of PER1/2 oscillations are further regulated by post-translational control of their stability and subcellular localization via CK1δ/ε phosphorylation (Vanselow *et al*, 2006; Gallego & Virshup, 2007; Kaasik *et al*, 2013; Qin *et al*, 2015; Zhou *et al*, 2015; Shinohara *et al*, 2017; Narasimamurthy *et al*, 2018; Ode & Ueda, 2018). Thus, CK1δ/ε is being considered as an effective pharmacologic target for the manipulation of circadian rhythms (Badura *et al*, 2007; Sprouse *et al*, 2009, 2010; Meng *et al*, 2010; Kim *et al*, 2013; Pilorz *et al*, 2014). The phase of circadian rhythms can also be adjusted by Per1/2 gene transcription induced by external light, which is transmitted from the retina to the hypothalamic suprachiasmatic nucleus (SCN) via the retinohypothalamic tract (Hattar *et al*, 2002; Gallego & Virshup, 2007). This leads to the entrainment of the endogenous circadian system to the external day–night cycle (Wright *et al*, 2013).

The failure of synchrony between the clock and external cycles can occur due to dysfunction of the circadian clock system or alteration of the external environment. Notably, recent epidemiological data suggest that more than 80% of the population appears to live a shift work lifestyle (Sulli *et al*, 2018). This increases the risk for various chronic diseases such as sleep disorders, cancer, diabetes, and mood disorders (Zhu & Zee, 2012; Sulli *et al*, 2018). To restore normal circadian phase, several approaches are being used. The

1 Department of Mathematical Sciences, Korea Advanced Institute of Science and Technology, Daejeon, Korea
2 Clinical Pharmacology, Pfizer Global Product Development, Pfizer Inc., Groton, CT, USA
3 Comparative Medicine, Worldwide Research & Development, Pfizer Inc., Cambridge, MA, USA
4 Enzymology and Transporter Group, Pharmacokinetics, Dynamics and Metabolism, Worldwide Research & Development, Pfizer Inc., Groton, CT, USA
5 Clinical Pharmacology and Pharmacometrics, Research & Development, Biogen Inc., Cambridge, MA, USA
6 Neuroscience Research Unit, Worldwide Research & Development, Pfizer Inc., Boston, MA, USA
7 Department of Animal Medicine, University of Massachusetts Medical School, Worcester, MA, USA
*Corresponding author. Tel: +1 8604411926; Fax: +1 8607159738; E-mail: cheng.chang@pfizer.com
**Corresponding author. Tel: +82 423502736; E-mail: jaekkim@kaist.ac.kr

most common is timed light exposure, because light can advance and delay circadian phase depending on exposure time (Khalsa *et al*, 1997). However, as this therapy often requires bright exposure for up to 2 h daily at specific times, patients show poor compliance with it, highlighting the need for different approaches. Pharmacological phase adjustment of circadian rhythms is considered as an attractive alternative to timed light exposure (Skelton *et al*, 2015). For instance, administration of melatonin in the evening can advance circadian phase of patients with delayed sleep phase disorder (DSPD) (Zhu & Zee, 2012). However, although melatonin given in the morning can delay circadian rhythms in experimental condition (Burgess *et al*, 2010), its clinical efficacy for advanced sleep phase disorder (ASPD) has not been reported (Sack *et al*, 2007; Zhu & Zee, 2012). Furthermore, other clock-modulating drugs including dopamine partial agonists and GABA receptor modulators have only shown efficacy in treating DSPD but not ASPD (Ozaki *et al*, 1989; Regestein & Monk, 1995; Takaki & Ujike, 2014; Omori *et al*, 2018; Takeshima *et al*, 2018). Given the absence of drugs effectively delaying circadian phase, PF-670462 (PF-670), a potent and selective CK1δ/ε inhibitor currently in preclinical development, is the most promising candidate compound for treating ASPD as it induces a large phase delay regardless of dosing timing (Badura *et al*, 2007; Kim *et al*, 2013).

The effect of clock-modulating drugs dramatically changes upon external light exposure because light also alters the circadian phase. For instance, the phase shifts induced by PF-670 or melatonin are attenuated by light exposure (Sprouse *et al*, 2010; Kim *et al*, 2013; Nesbitt, 2018). The effect of clock-modulating drugs is also proposed to be affected by individual variations in the molecular cause of circadian disruption and photosensitivity (Holst *et al*, 2016; Keijzer *et al*, 2017). However, the interindividual variability in pharmacological circadian phase shift caused by differences in individual genes and environmental lighting conditions has not been systematically investigated. Thus, whether precision medicine could be applied as successfully to circadian disruption as it has been to other areas of medicine such as oncology is currently unclear (Skelton *et al*, 2015; Holst *et al*, 2016; Keijzer *et al*, 2017). Indeed, current therapeutics are mainly tailored around the patient's clinical features (i.e., phase of sleep–wake cycle) (Zhu & Zee, 2012).

The SCN of diurnal and nocturnal mammals have similar structures and temporal patterns of the core clock gene expression and neuronal firing rate (Inouye & Kawamura, 1982; Cohen *et al*, 2010; Kumar, 2017; Millius & Ueda, 2018; Mure *et al*, 2018). Thus, pharmacomanipulation of the circadian clock has been expected to be similar in *nocturnal* and *diurnal* mammals. However, they display nearly antiphase rhythms in physiology and behavior (Refinetti, 2006). This suggests that differences may still exist in their responses to clock-modulating drugs, complicating the translation of preclinical studies into effective therapies.

We used the response to the selective CK1δ/ε inhibitor, PF-670, to dissect any differences in circadian modulation between diurnal and nocturnal mammals that may be clinically relevant. We found that nocturnal mice and a diurnal non-human primate (NHP), the cynomolgus monkey (*Macaca facicularis*), show a quantitatively different response: PF-670 induces a significantly smaller phase delay in NHPs than in mice. This interspecies variation was analyzed by developing a comprehensive systems chronopharmacology model based on experimental data from the NHPs and humans. The model

simulations for the intracellular interactions of PF-670 with clock components and follow-up experiments revealed that the weak drug response of NHPs is mainly due to the strong attenuation of PF-670-induced phase delay by light exposure, which appears to exist across many diurnal mammals including humans. This explains the absence of effective pharmacotherapy for delaying circadian phase of sleep disorder patients in a daylight cycle (Sack *et al*, 2007; Zhu & Zee, 2012). Our work reveals a previously unrecognized biological variable in translating the efficacy of clock-modulating drugs from nocturnal mice to diurnal humans: their different photosensitivity. Besides photosensitivity, we found that the effect of PF-670 on the circadian phase dramatically decreases as endogenous levels of PER2 decrease. Together with this finding, our *in silico* experiments revealed that depending on the molecular cause of circadian disruption and the patient's light exposure, which alters PER2 levels, the effect of the CK1δ/ε inhibitor (CK1i) changes. To overcome the inter- and intraindividual variations, we developed an adaptive chronotherapeutics that identifies a precise dosing time for CK1i to restore the normal circadian phase. Our study illustrates how a systematic approach can both identify sources of variations in drug response and generate a strategy providing a precise dosing regimen for circadian disruption.

# Results

## The effect of PF-670 is more strongly attenuated by light in diurnal NHPs than in nocturnal mice

PER1/2 phosphorylation by CK1δ/ε regulates their degradation, binding to CRY1/2, and nuclear translocation, which are the key processes of the transcriptional–translational negative feedback loop of the mammalian circadian clock (Fig 1Ai; Gallego & Virshup, 2007). When CK1δ/ε is inhibited by PF-670, these processes are slowed down (Fig 1Ai) and circadian phase is delayed, which is attenuated by light, the strongest zeitgeber (Fig 1Aii) (Badura *et al*, 2007; Walton *et al*, 2009; Meng *et al*, 2010; Sprouse *et al*, 2010; Kim *et al*, 2013). To analyze the effect of PF-670 on the circadian phase in diurnal NHPs, and compare it with nocturnal mice, we first compared the free PF-670 brain concentrations across species (see Materials and Methods for details). Despite the lower dose level in NHPs (10 mg/kg (mpk)) than used in our previous study in mice (32 mpk), the drug exposure in NHPs (AUC = 3.6 μM·h) was much higher than in mice (AUC = 0.5 μM·h) (Fig 1B; Kim *et al*, 2013). Due to the higher drug exposure in NHPs, we hypothesized that PF-670 induces a larger phase delay of activity onset in NHPs than in mice. To investigate this, we compared the phase delays of NHPs induced by 3-day 10 mpk dosing in a dark–dark cycle (DD) (Fig 1C) with the phase delays of mice dosed with 32 mpk PF-670 for 3 days in DD (Kim *et al*, 2013). Indeed, NHPs showed a significantly larger phase delay (5.2 h) compared with mice (3.8 h) (*P* = 0.03; Fig 1E) (see Materials and Methods for details of phase delay measurement). This larger phase delay in NHPs than in mice might be due to the different dosing times for NHPs (e.g., CT14) than for mice (CT11) as the effect of PF-670 changes upon dosing time (Badura *et al*, 2007). However, the phase delay of NHP induced by dosing at CT11 is also likely to be larger than that of mice because the dosing at CT11 is expected to yield a nearly maximal phase delay (Badura

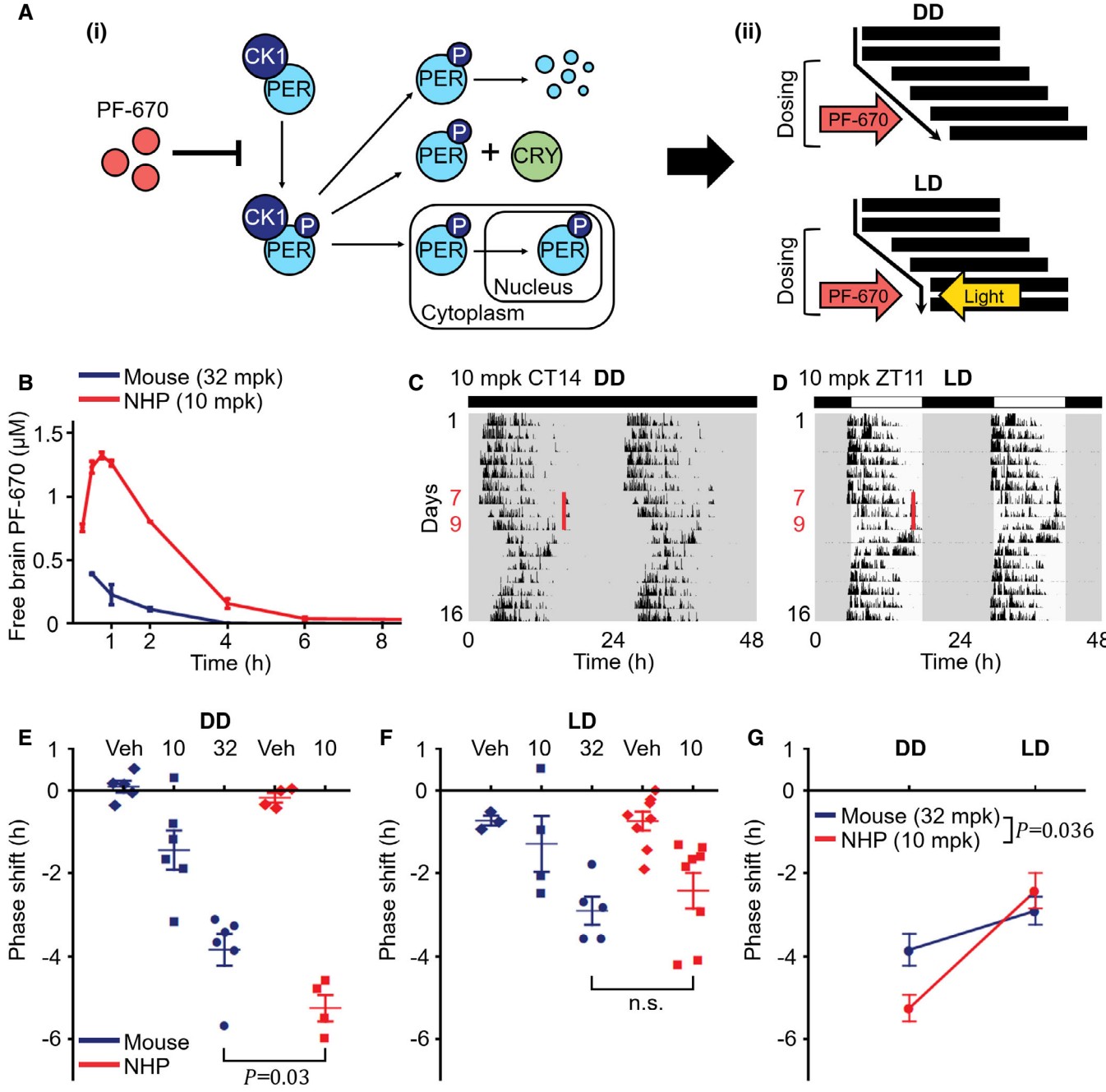

**Figure 1. Light attenuates the effect of PF-670 more strongly in diurnal NHPs than in nocturnal mice.**

A PF-670 inhibits the phosphorylation of PER by CK1δ/ε (i) and delays the circadian phase, which is counterbalanced by light (ii). Thus, daily dosing leads to continually accumulating phase delay in DD and constant stable phase delay in LD.

B Measured free PF-670 exposure in the brain tissue of mice and NHPs. Drug exposure in NHPs after administration of 10 mpk PF-670 (AUC = 3.6 μM·h) is ~7-fold higher than that in mice given 32 mpk PF-670 (AUC = 0.5 μM·h) (n = 2–3; mean ± SEM). The mouse data are adopted from (Kim *et al*, 2013).

C, D Double-plotted actograms of NHPs' activity for 16 days. NHPs were treated with 10 mpk PF-670 for 3 consecutive days at the same solar time of day: 4 pm for the DD experiment (C) and 4:30 pm (ZT11) for the LD experiment (D), respectively, which are highlighted as red lines. White and black rectangles indicate the times of light going on and off (LD 12:12).

E, F Phase delay induced by the 3-day dosing under DD (E) or LD (F). The phase of activity onset in NHPs (10 mpk) is more delayed than that in mice (32 mpk) under DD (E), but not under LD (F) (n = 3–8; P = 0.03, one-way analysis of variance (ANOVA); n.s., no significant difference). Veh denotes vehicle. The error bars represent mean ± SEM. The mouse data are adopted from (Kim *et al*, 2013).

G The quantification of (E) and (F) indicates that light has a stronger attenuating effect on the PF-670-induced phase delay in NHPs than in mice (n = 5–8; mean ± SEM; P = 0.036, two-way ANOVA).

et al, 2007) and the drug exposure is much higher in NHPs than in mice (Fig 1B) (see Fig EV3 after reading below two sections for details).

Next, we investigated whether there is also an interspecies difference in the alteration of PF-670-induced phase delay by light. Specifically, we compared the phase delays of NHPs induced by 10 mpk dosing for 3 days at zeitgeber time (ZT) 11 in a light–dark cycle (LD) (Fig 1D) with the previously reported ones of mice dosed with 32 mpk PF-670 under the same conditions (Kim et al, 2013). Despite the stronger effect of PF-670 in NHPs than in mice under DD (Fig 1E), NHPs did not show a larger phase delay than mice under LD (n.s.; Fig 1F). Thus, light attenuates the PF-670-induced phase delay more strongly in NHPs than in mice (P = 0.036; Fig 1G).

### A systems chronopharmacology model is developed to simulate the effects of PF-670 and light on circadian rhythms of NHPs

NHPs and mice show large differences in the pharmacokinetics of PF-670 (Fig 1B), and the effect of PF-670 on circadian phase, and how this is influenced by light (Fig 1E–G). To analyze such multiple differences systematically, we developed the first systems chronopharmacology model for NHPs by modifying our previous model (Kim et al, 2013), which successfully simulates the effects of PF-670 and light on the intracellular circadian clock of mice. The modified parts of the model including the newly estimated pharmacokinetic parameters and new equations for the light module are described in the Materials and Methods. See Appendix Equation S1, Tables EV1 and EV2, and Datasets EV1 and EV2 for the detailed description for the equations, variables, and parameters of the NHP model. In the NHP model, inhibition of CK1δ/ε for PER1/2 phosphorylation by PF-670 (Fig 2Ai) and light-induced Per1/2 gene transcription via CREB (Fig 2Aii) were incorporated to simulate the resulting phase shift of the circadian clock at the molecular level (orange arrow; Fig 2Aiii).

To match the simulated phase shift with the experimentally observed one in NHPs, firstly in the absence of light, we adjusted the model parameters describing the pharmacokinetics of PF-670 in NHPs (Fig 2Ai) (see Fig EV1A–C, Dataset EV1 and Materials and Methods for details). However, unlike DD dosing, the effect of LD dosing in NHPs (Fig 1F) was not captured by the model (Fig EV1D and E), necessitating revision of the light module. In the previous mouse model, light was assumed to equally promote Per1/2 gene transcription across the entire circadian phase for simplicity (Kim et al, 2013). However, the light-induced Per1/2 gene transcription is known to be suppressed more strongly during the subjective day than during the subjective night (Albrecht et al, 1997; Shigeyoshi et al, 1997; Miyake et al, 2000; Kiessling et al, 2014). Therefore, this gating for light, which reduces photosensitivity of the circadian clock depending on the circadian time (CT) (Geier et al, 2005), was incorporated into the light module of the new model (Fig 2Aii). Furthermore, adaptation for light, which reduces photosensitivity depending on light duration (Ding et al, 1997; Jagannath et al, 2013; Cao et al, 2015), was also added to further improve the accuracy of the light module (Fig 2Aii).

The gating and adaptation, which have not been experimentally quantified in NHPs, were estimated by fitting the model to the LD dosing effect that we observed in NHPs (Fig 2B) and human light phase response curves (PRCs) (Khalsa et al, 1997, 2003) (see

Materials and Methods for details). The shapes of the gating and adaptation (Fig EV1F and H) were randomly chosen during the estimation process (Fig EV2A–E), However, despite that, we found that all the identified pairs of gating and adaptation had a similar shape (Figs 2Aii and EV2F). Specifically, the identified gatings commonly yielded high photosensitivity during the subjective night, which is consistent with experimental data from other mammals (Albrecht et al, 1997; Shigeyoshi et al, 1997; Miyake et al, 2000; Kiessling et al, 2014). With the identified 10 adaptations, light-induced Per1/2 gene transcription gradually decreased and reached a 50% reduction after ~8 h.

### The new light module accurately predicts the response of NHPs to light exposure

As the new light module of the model was estimated mainly based on the response of humans to light (Fig EV2), we next investigated whether it could accurately predict the potent light-induced attenuation of the PF-670 effect in NHPs. To solely focus on the effect of light, we introduced a 3-day 2-h delayed LD on days 8–10, which mimics the ~2-h phase delay of NHPs induced by the 3-day LD dosing (Fig 2C). After this 3-day "jet lag", the delayed phase was advanced to the original circadian phase for days 11–13. We found that such light-induced phase shifts that occur during and after jet lag were accurately predicted by the model with the new light module (Fig 2D). This indicates that the new light module precisely captures the light-induced phase advance in NHPs against the PF-670-induced phase delay in LD dosing.

We next investigated whether the model can predict the combined effect of PF-670 and light even when the dosing time changes. We chose dosing at ZT4 as it leads to a much weaker phase delay in mice than dosing at ZT11 (Badura et al, 2007; Kim et al, 2013). However, in NHPs, dosing at ZT4 led to a similar phase delay as dosing at ZT11 (1.5; ZT4 and 1.7 h; ZT11; Fig 2B, E and F). This unexpected drug effect in NHPs was accurately predicted by the model with the new light module (Fig 2F), which is mainly due to the strong attenuation of the drug effect by light in NHPs (Fig EV3). Taken together, the difference in light response between mice and NHPs is a critical factor leading to their heterogeneous response to a clock-modulating drug.

### The strong attenuating effect of light is expected in humans

Although the new light module was developed mainly based on the human response to light (Fig EV2), the model accurately captured the effect of light in NHPs in both the presence and absence of dosing (Fig 2). Thus, the strong light-induced attenuation of the PF-670 effect in NHPs compared with mice (Figs 1G and 3A) is expected to also exist in humans. To investigate this, we used our previous finding that the magnitude of the advance zone of the 12-h light PRC can determine how strongly a 12-h light pulse attenuates the effect of PF-670 in LD 12:12 (Kim et al, 2013). Indeed, reflecting the stronger light-induced attenuation in NHPs than in mice (Figs 1G and 3A), the simulated PRC of NHPs had a larger magnitude of the advance zone than the experimentally measured one of mice (Fig 3B) (Comas et al, 2006). This leads to a higher ratio of maximum magnitudes between the advance (A) zone and the delay (D) zone (A/D ratio) of light PRC in NHPs (A/D = 1.25) than in

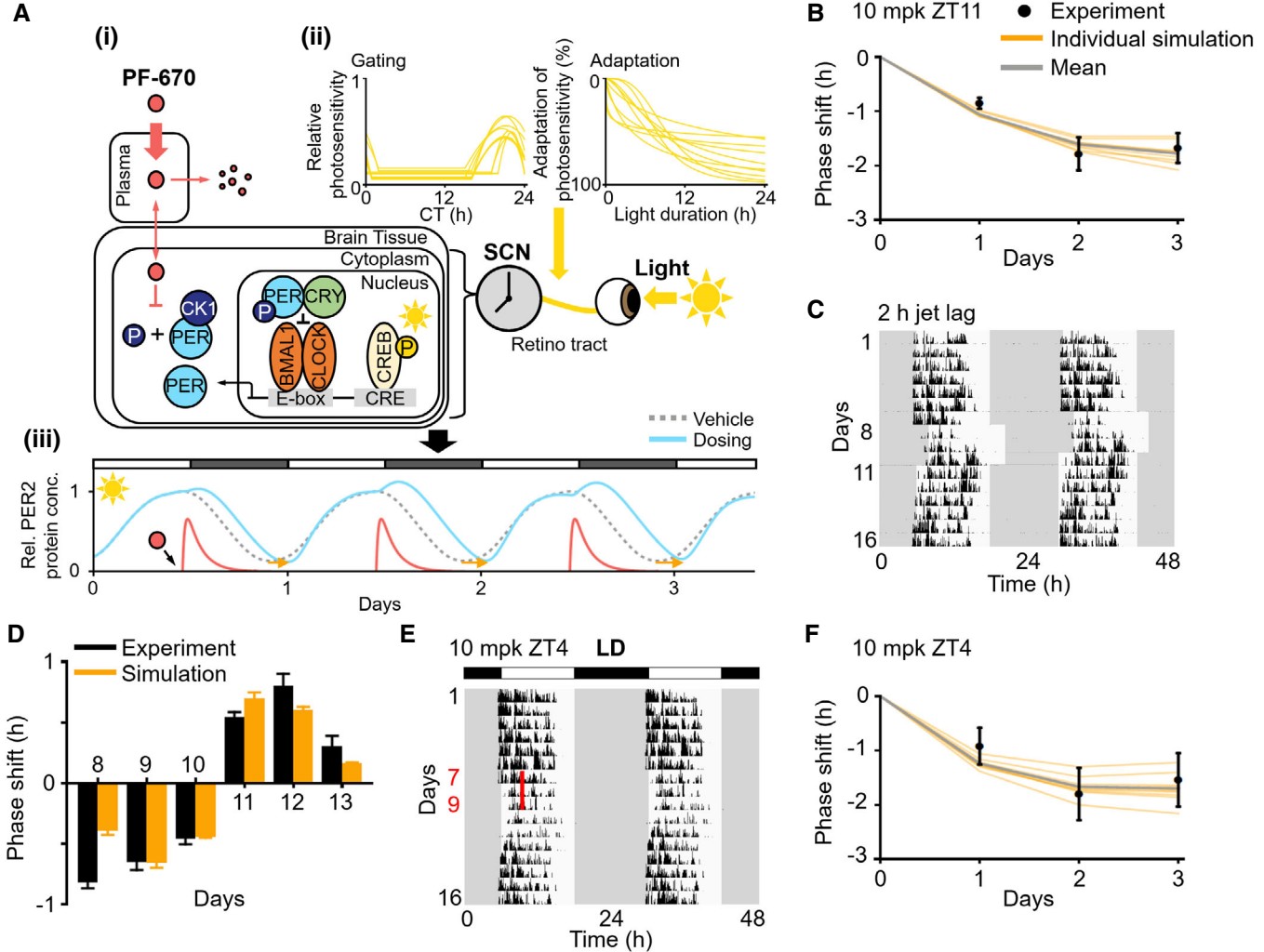

**Figure 2. Systems chronopharmacology model accurately simulates the phase shift of NHPs induced by PF-670 and light.**

A  Description of the systems chronopharmacology model for NHP. (i) The translocation of PF-670 from plasma to the SCN and its inhibition of PER phosphorylation by CK1δ/ε in NHP (Fig EV1A) were accurately modeled to reproduce the PF-670 exposure (Figs 1B and EV1B) and its effect in NHPs (Figs 1E and EV1C). (ii) The dependence of light-induced Per1/2 gene transcription via CREB on circadian time (gating) and light duration (adaptation) is included in the model (Fig EV1F–H). The estimated 10 pairs of gating and adaptation allow for the model to successfully reproduce the light-induced phase shift in NHPs and humans (Fig EV2). (iii) The model simulates the combined effect of PF-670 (i) and light (ii) on PER2 rhythms in the SCN of NHPs.

B  The model accurately reproduces the phase delay of NHPs induced by 3-day LD dosing at ZT11. Here, the PF-670-induced phase delay from which the vehicle-induced phase shift is subtracted was used (*n* = 8; mean ± SEM; Fig 1F). Each orange line is an individual simulation result obtained with each pair of gating and adaptation in (Aii), and their average is represented by the gray line.

C  Double-plotted actogram of NHPs' activity for 16 days. The light is on at 6 am and off at 6 pm except for days 8 to 10 when the light is on at 8 am and off at 8 pm (i.e., LD is delayed by 2 h).

D  The model successfully predicts the phase shift induced by 2-h jet lag (days 8–10) and its recovery after jet lag (days 11–13) (*n* = 7; mean ± SEM; C).

E  NHPs were treated with 10 mpk PF-670 for 3 consecutive days at ZT4 in LD 12:12, which are highlighted as the red line.

F  The phase delay induced by 3-day LD dosing at ZT4 (*n* = 8; mean ± SEM; E) is accurately predicted by the model.

mice (A/D = 0.41). This is consistent with experimental studies in which diurnal animals generally have PRCs with higher A/D ratios than nocturnal animals (Johnson, 1992). In particular, the human PRC has a higher A/D ratio than the mouse PRC for both short and long light pulses (Fig 3C) (Comas *et al*, 2006; St Hilaire *et al*, 2012; Ruger *et al*, 2013). This supports our proposal that the PF-670-induced phase delay is also strongly attenuated by external light in humans, similar to NHPs.

### The effect of CK1i depends on individual photosensitivity

Large interindividual variability of photosensitivity up to ~3-fold has been observed in humans (van der Meijden *et al*, 2016; Stone *et al*, 2018; Watson *et al*, 2018). Given the strong attenuating effect of light on CK1δ/ε inhibition in humans (Fig 3C), we would expect a potentially large variation in CK1i response in individuals with different levels of photosensitivity (Fig 3D). To investigate this, among

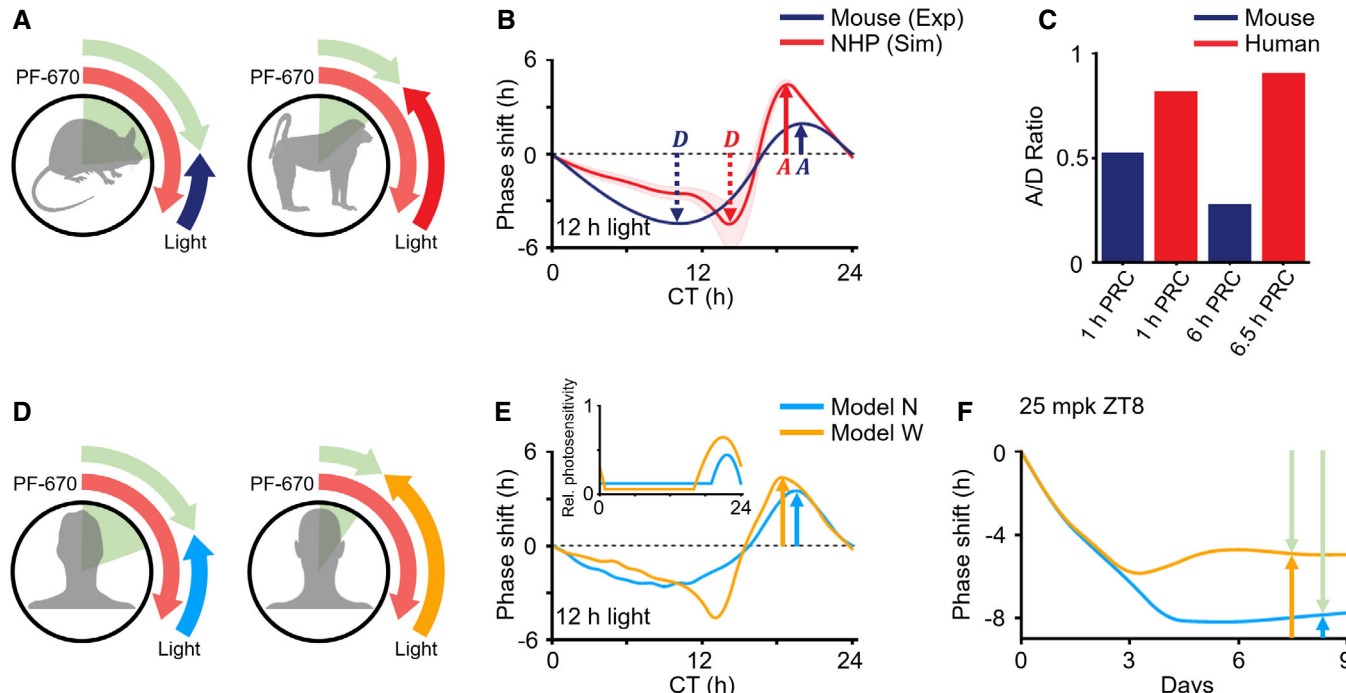

**Figure 3. Inter- and intraspecies variability in photosensitivity causes variation in the effect of PF-670.**

A    The stronger attenuating effect of light in diurnal NHPs than in nocturnal rodents leads to the interspecies variation observed in the response to PF-670.
B    The simulated PRCs of diurnal NHPs have a higher ratio of maximal magnitudes between advance and delay zones (A/D = 1.25) than that of the experimentally measured nocturnal mouse PRC (A/D = 0.41), which is adopted from (Comas *et al*, 2006). Here, the red line and red range represent the mean ± SD of the simulated 12-h light PRCs of the NHP model with different gating and adaptation (Fig 2Aii).
C    Similarly, the light PRC of diurnal humans has a higher A/D ratio (St Hilaire *et al*, 2012; Ruger *et al*, 2013) than that of nocturnal mice (Comas *et al*, 2006).
D    The effect of PF-670 can be heterogeneous due to interindividual variability in photosensitivity.
E    Model N with a narrow high photosensitivity zone (inset) simulates a smaller magnitude of the advance zone of 12-h light PRC than model W with a wide high photosensitivity zone.
F    Thus, model N simulates a larger constant stable phase delay than model W when a single daily 25 mpk dose is given at ZT8 under LD 12:12.

the 10 pairs of gating and adaptation (Fig 2Aii), two pairs were chosen: One has a narrower high photosensitivity zone than the other (Fig 3E inset) (see Fig EV4A and Dataset EV2 for details). Thus, the model with the narrow high photosensitivity zone (model N; Fig 3E) simulates the smaller magnitude of the advance zone of light PRC than the model with the wide high photosensitivity zone (model W; Fig 3E). Due to the weaker attenuating effect of light in model N than in model W (Fig 3E), PF-670 induced a larger phase delay in model N (Fig EV4B and C). For instance, when a single daily 25 mpk dose is given at ZT8, the PF-670-induced phase delay is balanced with the light-induced phase advance, and reaches the equilibrium point after 1 week of dosing (Fig 3F) (Kim *et al*, 2013). At this point, with a weaker attenuating effect of light in model N, the constant stable phase delay induced by PF-670 is larger than in model W. Furthermore, when a higher dose is used, the effect of CK1i under LD differs qualitatively depending on individual photosensitivity: The PF-670-induced phase delay overcomes the light-induced phase advance, and thus, it accumulates in model N while the phase shift unstably alternates between delay and advance in model W due to amplitude suppression (Fig EV4D–G) (Diekman & Bose, 2018). Thus, the degree of photosensitivity of the individual likely has dramatic consequences for the effect of clock-modulating drugs.

## The effect of CK1i becomes stronger as PER2 levels increase

Due to the larger phase delay induced by PF-670 compared to other clock-modulating drugs (Subramanian & Subbaraj, 1993; Badura *et al*, 2007; Burgess *et al*, 2010), a daily dosing of PF-670 can induce a stable phase delay even in the presence of daylight cycle (Fig 3F). This suggests that daily dosing of CK1i could be a promising first pharmacological candidate for ASPD. However, as the effect of PF-670 depends on the dose level and dosing time, those values need to be carefully selected. This raises the question of what constitutes a precise dosing regimen to treat ASPD. For instance, to treat ASPD patients with a 4 h more advanced circadian phase and sleep schedule than healthy people, one candidate regimen would be a single daily dosing, which is known to induce the ~4-h stable phase delay in healthy people. Alternatively, dosing time may need to be advanced by 4 h to reflect the advanced circadian phase of ASPD patients compared with healthy people, as is done in timed phototherapy (Bjorvatn & Pallesen, 2009; Zhu & Zee, 2012). Furthermore, dosing regimen could differ depending on the genetics and environmental lighting conditions of ASPD patients.

To investigate this, we performed a series of *in silico* experiments using the systems chronopharmacology model. We first identified

four dosing regimens that would induce the ~4-h stable phase delay in the WT model having normal circadian phase, which mimics healthy people: single daily 26 mpk, 24 mpk, 22 mpk, or 19 mpk dosing at ZT4, 5, 6, or 7, respectively (blue circles in WT; Fig 4B) (see Dataset EV2 for details of the WT model). Then, these dosing regimens are applied to eight different ASPD models, which we developed by incorporating various mutations including PER2 FASPS (Toh et al, 2001; Vanselow et al, 2006) into the WT model (see Dataset EV3 and Materials and Methods for details). These ASPD models simulate a shorter period than the WT model in DD and thus have a ~4-h more advanced circadian phase in LD (Fig 4A). Interestingly, the dosing regimens inducing the ~4-h phase delay in the WT model led to much larger phase delays in all of the ASPD models, with large variations (blue circles; Fig 4B). On the other hand, when the dosing times were advanced by 4 h reflecting the advanced circadian phase of the ASPD models, as is done in timed phototherapy (Bjorvatn & Pallesen, 2009; Zhu & Zee, 2012), the model simulated phase delays closer to the desired ~4-h phase delay (red squares; Fig 4B). Nevertheless, they were still larger. We found that this is mainly due to the weaker attenuating effect of light in the ASPD models than in the WT model (Fig EV5A). Importantly, despite the same clinical feature of the ASPD models (4-h advance) (Fig 4A), the PF-670-induced phase delay is considerably different (e.g., $Per2^t$ and $Cry1^t$; Fig 4B).

To identify the source of the heterogeneous drug response among the ASPD models, we estimated the relationship between the effect of PF-670 (red squares; Fig 4B) and the average level of various core clock molecules of the ASPD models (Fig EV5B). Interestingly, we found that the average protein level of PER2 is significantly more strongly correlated with the effect of PF-670 than that of the other clock proteins (Fig EV5C). Specifically, as the average PER2 protein levels (Fig 4A inset) increase in the ASPD models, the effect of PF-670 (red squares; Fig 4B) becomes stronger (Fig 4C). This correlation is not due to the different free-running period of the ASPD models (Fig EV5D) and appears to stem from the fact that phosphorylation of PER2 by CK1$\delta$/$\varepsilon$ is the target of PF-670. Indeed, when we increase PER2 level in the model by tuning model parameters, the effect of CK1i becomes stronger regardless of its effect on the free-running period (Fig EV5E and F). Furthermore, PER2 abundance also explains the different effect of PF-670 depending on day length: Due to the higher PER2 abundance at the dosing times in LD 8:16 than in LD 16:8 (Fig EV5G), a larger PF-670-induced phase delay is simulated in LD 8:16 than in LD 16:8 (Fig 4D). To support these *in silico* predictions (Fig 4C and D), we estimated the relationship between the effect of PF-670 and PER2 levels from experimentally measured PRC to PF-670 (Badura et al, 2007) and time series of PER2 levels in the SCN (Amir et al, 2004). Indeed, we found a strong positive correlation between them, which is also recapitulated by our model (Fig 4E).

### Adaptive chronotherapeutic approach identifies precise dosing times

We found that the molecular cause of ASPD and the patient's environmental lighting conditions dramatically influence PER2 levels and thus the effect of CK1i (Fig 4C and D). This dependence of the drug effect on PER2 level could be used to identify a precise dosing regimen. That is, due to the accumulation of PER2 during the daytime (Fig 4A), the drug effect becomes stronger or weaker when

the dosing time of day is delayed or advanced, respectively (Fig EV4B and C) (Badura et al, 2007; Kim et al, 2013). By using this feature, we developed an adaptive chronotherapeutic approach: If the current dosing regimen leads to a weaker or stronger drug effect than the desired one, the dosing time is delayed or advanced by 1 h, respectively, until the desired phase delay is achieved (Figs 5A and EV6A).

To test whether this approach works as expected despite the large perturbation of the circadian clock (e.g., PER2 abundance and phase) by genetic variation or environmental lighting conditions, we applied the adaptive chronotherapeutic approach to all ASPD models with varying day lengths. Specifically, a single daily 10 mpk dose was given at ZT3 to the ASPD models in LD 16:8 (day 1; Fig 5B). Then, depending on the induced phase delay, the initial dosing time (ZT3) was adjusted according to the chronotherapeutics (Figs 5A and EV6A). This identified the precise dosing time (colored number; Fig 5B) inducing the desired phase delay for all ASPD models regardless of their molecular cause (gray range; Fig 5C). This chronotherapeutic approach also successfully identified the precise dosing time when day length changes to LD 8:16 (Fig 5D and E). Due to the stronger drug effect in LD 8:16 than in LD 16:8 (Fig 4D), the identified precise dosing time was more advanced in LD 8:16 (Fig 5D) than in LD 16:8 (Fig 5B). Interestingly, even if ASPD models have the same precise dosing time in LD 16:8 (e.g., ZT3 for $Per2^P$ and $Per2^n$; Fig 5B), their precise dosing times in LD 8:16 can be different (e.g., ZT3 for $Per2^P$ and ZT23 for $Per2^n$; Fig 5D). This further emphasizes the need for an adaptive chronotherapeutic approach to reflect the patient's environmental lighting condition.

Although solely adjusting the dosing time was successful (Fig 5B–E), it took longer to achieve the desired phase delay (e.g., 18 days for $Cry2^t$ in LD 8:16; Fig 5D). Furthermore, the identified precise dosing time could be one at which it is inconvenient to take a drug (e.g., ZT19 for $Cry2^t$; Fig 5D). This problem can be resolved if dose level and dosing time are adjusted simultaneously (Fig EV6B–F).

## Discussion

We unexpectedly found that the effect of CK1i on circadian phase differs between nocturnal mice and diurnal NHPs due to their different levels of photosensitivity: Light attenuates the effect of CK1i more strongly in NHPs than in mice. Such a strong attenuating effect of light on a drug-induced phase delay is expected to exist in humans as their light PRC has a large magnitude of the advance zone (St Hilaire et al, 2012; Ruger et al, 2013). Thus, the phase delay induced by melatonin would also be expected to be strongly attenuated by light. This would explain its weak effectiveness in treating ASPD and jetlag after westward travel in a daylight cycle (Sack et al, 2007; Zhu & Zee, 2012; Spiegelhalder et al, 2017). These results indicate that such interspecies difference in photosensitivity should be considered when translating the efficacy of clock-modulating drugs from nocturnal mice to diurnal humans.

Our systems chronopharmacology model based on experimental data from NHPs and humans predicts that a regimen reflecting a patient's individual circadian phase can achieve a more precise

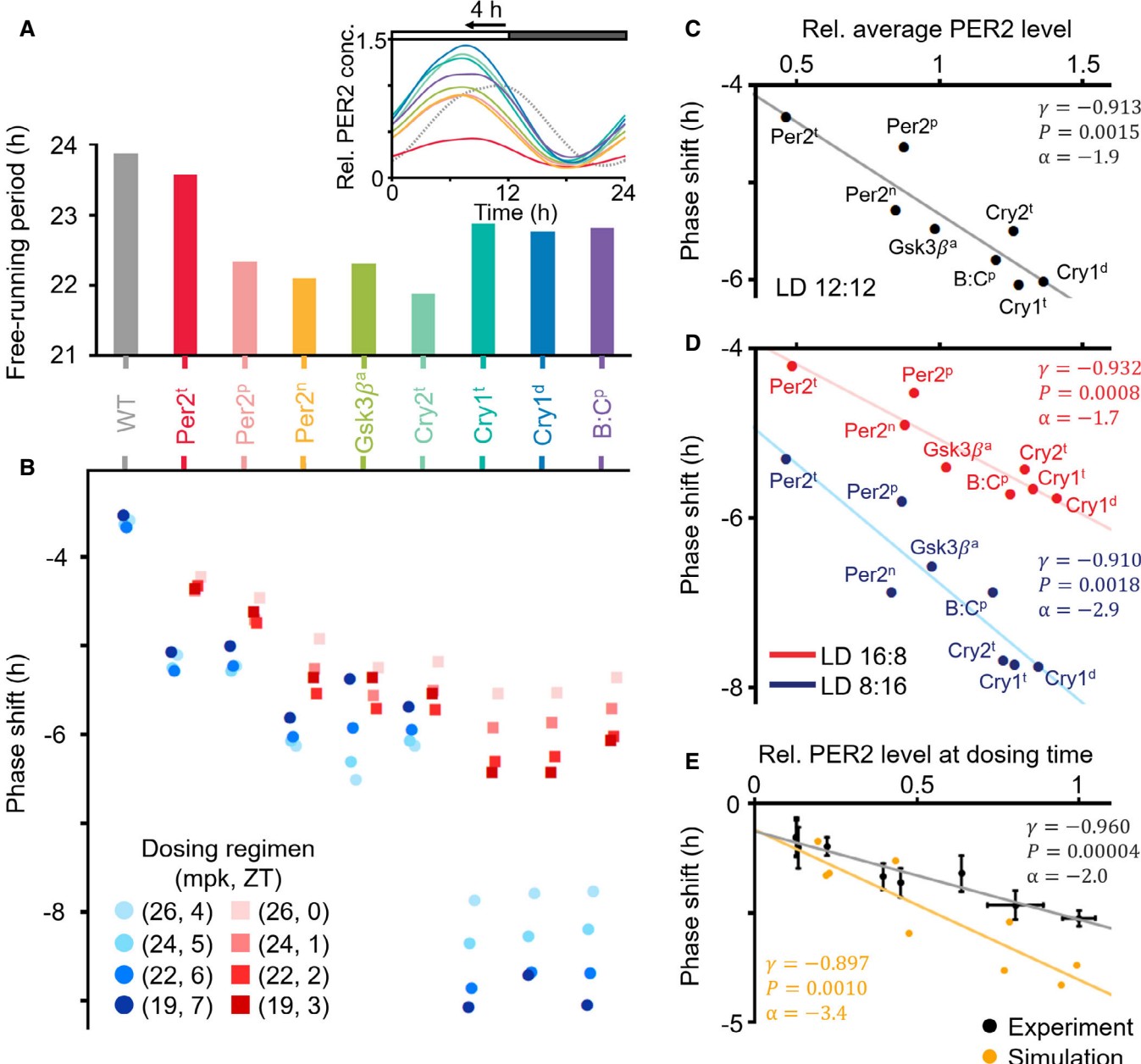

**Figure 4. CK1i induces a larger phase delay as PER2 levels increase depending on the molecular cause of ASPD and external lighting conditions.**

A   The simulated period of ASPD models is shorter than that of the WT model in DD. Thus, they have a ~4 h more advanced phase than the WT model in LD 12:12 (inset). PER2 protein concentrations in each mutant are normalized to the maximum PER2 concentration of the WT model (dashed gray line; inset).

B   The phase delays induced by a single daily dosing of PF-670 were simulated. The dosing regimens inducing ~4-h phase delays in the WT model led to much larger phase delays in all ASPD models with large inter- and intravariations (blue circles). When the dosing times are advanced by 4 h reflecting the advanced phase of the ASPD models, the phase delays become closer to but still larger than the ~4-h phase delays (red squares).

C   As average PER2 level is higher in the ASPD models (A), PF-670 induces a larger phase delay. Here, the phase shift is the average of the phase delays represented as the red squares in (B). The line represents the least-square fitting line. r and P denote the Pearson's correlation coefficient and P-value of Pearson's correlation test, respectively. α denotes the slope of the least-square fitting line.

D   The ASPD models simulate the larger PF-670-induced phase delay in LD 8:16 than in LD 16:8 due to the higher PER2 level in LD 8:16 than in LD 16:8 when dosing occurs (Fig EV5G).

E   Consistently, higher experimentally measured PER2 levels in the SCN (n = 4; mean ± SEM; Amir et al, 2004) at the dosing time leads to a larger PF-670-induced phase delay (n = 3–20; mean ± SEM; Badura et al, 2007), which was also captured by the NHP model when 20 mpk dosing was used.

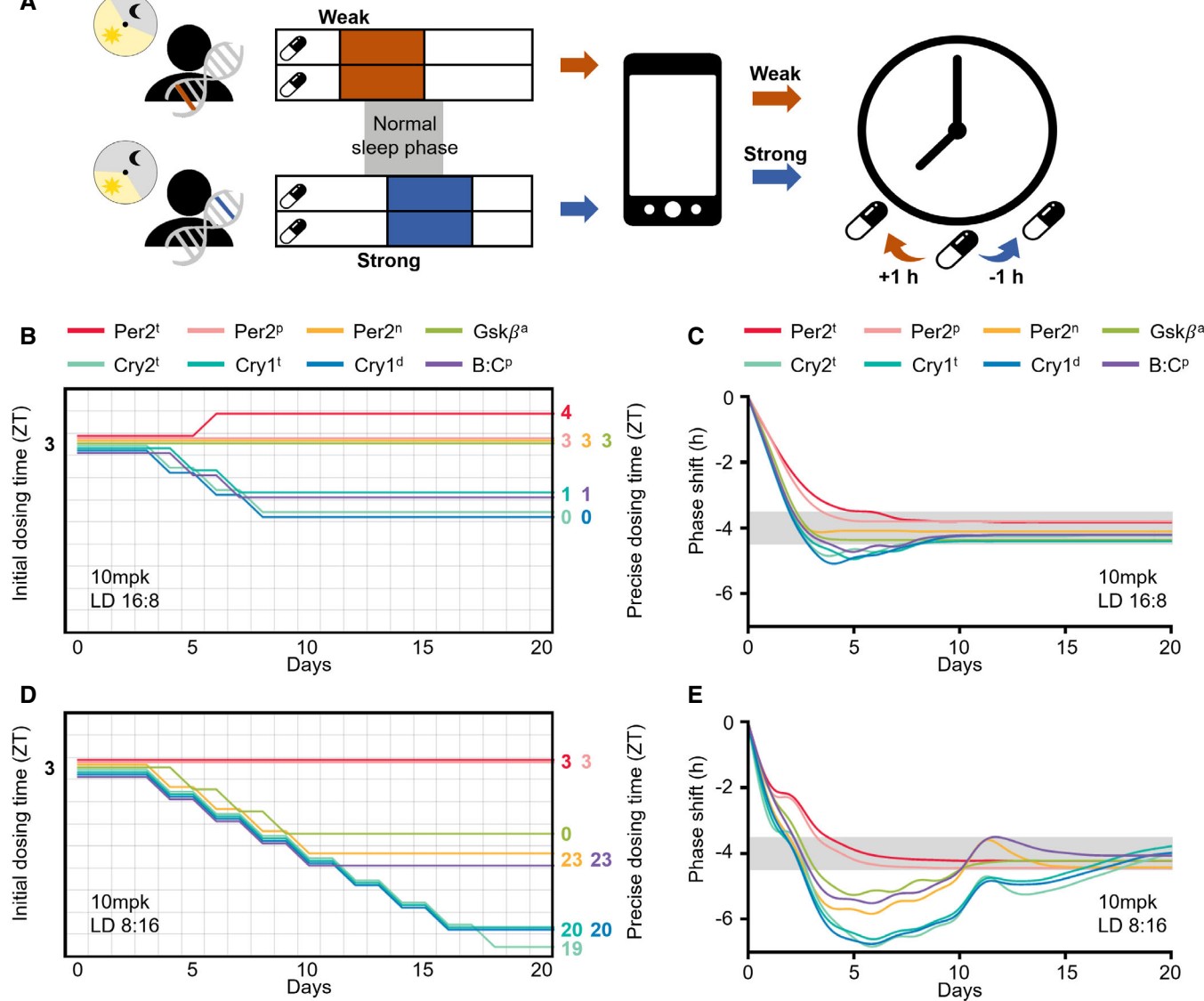

**Figure 5. An adaptive chronotherapeutic approach identifies the precise dosing regimen.**

A By tracking the patients' heterogeneous drug response caused by different genetics and environments, adaptive chronotherapeutics adjusts the dosing time: If the current dosing regimen leads to a weaker or stronger drug effect than the desired one, the dosing time is delayed or advanced by 1 h, respectively (see Fig EV6A for details).

B, C By adjusting the initial dosing time (ZT3) according to the adaptive chronotherapeutics, the precise dosing time can be obtained for each ASPD model (colored number; B), with which a single daily 10 mpk dosing leads to the desired ~4-h phase delay in LD 16:8 (gray range; C).

D, E When day length decreases to 8 h (LD 8:16), the precise dosing times are considerably advanced.

therapeutic effect than a non-circadian-based regimen. This may explain why the current circadian-based treatment can alleviate sleep disorders (Zhu & Zee, 2012; Sletten *et al*, 2018). However, our model predicts that even if the circadian phase of patients is similar, their response to CK1i can be considerably different depending on the molecular cause and the environmental lighting condition (Fig 4A–D). We found that such variability in the CK1i effect is mainly due to altered PER2 abundances (Figs 4C–E and EV5B–F). While the interspecies difference in PER2 abundance has not been investigated, it may contribute to the interspecies variability in the

CK1i effect (Figs 1E–G and 3A). Furthermore, the interspecies difference in the phase of PER2 rhythms (Vosko *et al*, 2009; Zhang *et al*, 2014; Millius & Ueda, 2018; Mure *et al*, 2018), which leads to the interspecies difference in PER2 abundance when dosing occurs, could also be the source of the interspecies variability in the CK1i effect. It would be interesting to investigate such a relationship between the effect of CK1i and PER2 abundance, and extend this to other clock-modulating drugs and their target molecules (e.g., KL001 targeting CRY; Hirota *et al*, 2012). Furthermore, as PER2 regulates the stability of p53, a key tumor suppressor (Gotoh *et al*,

2016), altered PER2 levels depending on genetics and environmental lighting conditions could vary the level of p53 as well. This may also explain the large inter- and intrapatient variability in the chronotherapy of antitumor drugs (Altinok *et al*, 2009; Levi *et al*, 2010).

Our work indicates that patient-specific molecular and lifestyle information should be integrated to achieve a desired therapeutic effect of clock-modulating drugs (Fig 4B–D). However, because obtaining such information can be challenging, we developed an adaptive chronotherapeutics, which identifies the precise dosing time to achieve normal circadian phase by tracking the patient's drug response (Figs 5A and EV6A and B). This adaptive chronotherapeutics requires a precise quantification of the drug-induced circadian phase shift in daily life. Recently, to overcome the insufficient accuracy of actigraphy and labor-intensive measurement of dim light melatonin onset (Duffy & Dijk, 2002; Ancoli-Israel *et al*, 2003), various methods have been developed, which are based on the metabolite timetable (Kasukawa *et al*, 2012), one-time phase estimation (Wittenbrink *et al*, 2018), or sleep-tracking using mobile phone (Walch *et al*, 2016). With these advances, the adaptive chronotherapeutics could play a critical role in incorporating biomarker data and providing real-time patient-tailored chronotherapy via smart devices (i.e., digital medicine) (Elenko *et al*, 2015).

# Materials and Methods

## Reagents and Tools table

| Reagent/Resource | Reference or Source | Identifier or catalog number |
|---|---|---|
| **Experimental models** | | |
| Mauritian Cynomolgus Monkey (Macaca fascicularis) | Charles River BRF, Inc. | |
| PF-670462 | Pfizer Medicinal Chemistry, Groton, Connecticut USA | |
| Plasma protein binding | Biopharma Drug Disp 23: 327–338 | |
| Brain tissue binding | J Pharm Sci 105: 965–971 | |
| **Chemicals, enzymes, and other reagents** | | |
| Sulfobutyl ether β-cyclodextrin | Sigma-Aldrich | CAT 1065550 |
| Control monkey plasma | Bioreclamation www.bioivt.com | |
| Acetonitrile | Fisher Chemical | CAT A998 |
| Ammonium formate | Sigma-Aldrich | CAT 78314 |
| Acetic acid | Fisher Chemical | A38-S |
| Ammonium acetate | Sigma-Aldrich | A7262 |
| Isopropyl alcohol | Fisher Chemical | A461-1 |
| **Software** | | |
| Analyst Software v 1.4.2 | AB Sciex, Inc. https://sciex.com/products/software/analyst-software | |
| Actical Software | Philips Respironics, Bend, Oregon | |
| ClockLab v6.0.52 | Actimetrics, Inc. Wilmette, IL | |
| Wolfram mathematica | Wolfram Research, Inc. https://wolfram.com | |
| GraphPad Quickcalcs software | GraphPad Software, Inc. http://www.graphpad.com/quickcalcs/index.cfm | |
| **Other** | | |
| Actical activity monitor | Philips Respironics, Bend, Oregon | Model IPX7 |
| ActiReader | Philips Respironics, Bend, Oregon | Model 1063543 |
| Actical animal armored primate collar | Philips Respironics, Bend, Oregon | CAT 198-0032-00M |
| Digital light meter | Amprobe, Inc. | Model LM631A |
| Purina Lab Diet | W.F. Fisher and Son | CAT 5K91 |
| Vacutainer spray-coated K2EDTA | Becton Dickinson Company | CAT 367835 |

**Reagents and Tools table** (continued)

| Reagent/Resource | Reference or Source | Identifier or catalog number |
|---|---|---|
| AB Sciex API4000 tandem quadrupole mass spectrometer | AB Sciex, Inc. https://sciex.com/products/mass-spectrometers/triple-quad-systems/api-4000-system | |
| Shimadzu LC20AD pumps | Shimadzu Scientific Instruments https://www.ssi.shimadzu.com/products/liquid-chromatography/lc-20ad.html | |
| CTC PAL autosampler | Leap Technologies https://www.leaptec.com/collections/instruments/products/pal3 | |
| Synergi Max RP UPLC analytical liquid chromatography column 4 μ, 80 Å, 2 × 30 mm | Phenomenex | Part 00A-4337-B0 |

## Methods and Protocols

### Behavioral studies
#### Animals
All procedures involving animals were conducted with the approval of the Pfizer IACUC and were compliant with the *Guide for the Care and Use of Laboratory Animals* and the regulations and standards of the Animal Welfare Act (9CFR2, 9CFR3). Eight 5- to 6-year-old male Mauritian Cynomolgus macaques (Charles River BRF, Inc.) SPF for CHV-1, SRV 1, SRV 2, and SRV 5, STLV1, and SIV were used for behavioral studies. Six animals were pair-housed and two singly housed. All monkeys were chair-trained and fitted with collar-mounted Actical® activity monitors (Philips Respironics, Bend, Oregon, USA). The Actical® Software (Respironics ActiReader) was set to record activity counts in 5-min bins. ClockLab 6.0.52 (Actimetrics Wilmette, IL) was used to generate actograms and to determine activity onset and free-running periods. Animals were acclimated and housed in a room under 12:45/11:15 LD with a 45-min ramping period to full light on/off as simulated dawn/dusk (i.e., at 5:15 am lights started ramping up to full lights on at 6:00 am; at 5:15 pm lights started ramping down to completely off at 6:00 pm). The direct light intensity in the room was 1,300 lux at 1 meter from the floor in the middle of the room, which is measured using a digital light meter (Model LM631A, Amprobe, Everett, WA). For constant dim light conditions, the light intensity was ~14 lux measured at the cage face. Animals were placed under dim light conditions for 2 weeks prior to study onset and were under dim light conditions for 4 months during the study. Animals were fed Purina LabDiet 5K91 (W.F. Fisher and Son) twice daily, and reverse osmosis purified municipal water was provided *ad libitum*. Daily environmental enrichment was provided in the form of fruit treats and manipulada. Animals were chair-restrained for drug administration and data download from the Actical® monitors.

The sample size for behavioral studies was determined to detect the desired magnitude of difference (≥ 20 min), and the variance of the estimates was based on previous data where available and setting α = 0.05 and 1-β = 0.9. One animal was identified as an outlier in a 2-h jet lag experiment (Fig 2C) by Grubbs' test (α = 0.01) and was thus excluded from the analysis.

### Study design
A cross-over design was used for each treatment limb, and animals were randomized into 2 groups of 4 prior to each dosing limb.

PF-670462 was dissolved in 20% w/v sulfobutyl ether β-cyclodextrin, and filter sterilized. Formulations were prepared 1–2 h prior to administration and given subcutaneously for 3 consecutive days at solar time: 4:00 pm in DD and 4:30 pm in LD. Prior to DD dosing (Fig 1C), 4 NHPs were placed in DD for 2 weeks to estimate their free-running period. Their different intrinsic periods (23.8, 24, 24.1, and 24.3 h) caused them to be initially dosed at different CTs (CT4, 6, 8, and 14 h) since the CT when the initial dosing occurs was calculated by considering the activity onset at the pre-dosing day as CT0. Prior to LD dosing (Figs 1D and 2E), NHPs were placed in LD for at least 10 days for a stable base line (i.e., entrainment). This allowed for them to be initially dosed at the same ZT11. For the light-induced phase shift experiment (Fig 2C), following 7 days of stable activity, the time for lights on/off was delayed 2 h for 3 days and then returned to colony normal.

### Pharmacokinetic studies
#### Distribution profiles
PF-670462 was synthesized and characterized by Medicinal Chemistry at Pfizer Worldwide Research and Development (WRD). Animal studies were performed in accordance with the *Guide for the Care and Use of Laboratory Animals* (Clark *et al*, 1996) using protocols reviewed and approved by the WRD IACUC.

1. A single dose (10 mpk, SC) of PF-670462 was prepared in 20% sulfobutyl ether β-cyclodextrin in water (10 mg/ml).
2. The dose was administered (1 ml/kg) to male cynomolgus macaque monkeys by subcutaneous route ($N = 2$).
3. Serial blood samples were collected from the femoral vein at 0, 0.25, 0.5, 0.75, 1, 2, 4, 6, and 24 h post-dose into Vacutainer (Becton Dickinson Company, Franklin Lakes, NJ) blood collection tubes pretreated with EDTA and placed immediately on ice.
4. The whole blood samples were centrifuged for 10 min at 2,300 × g and plasma transferred to clean tubes for storage at −20°C.

Monkey pharmacokinetic samples were prepared and analyzed using a non-validated bioanalytical assay. For standard curve preparation, the plasma was purchased from Bioreclamation Inc., Hicksville, NY.

5. The standard curves were prepared in control cynomolgus monkey plasma via serial dilution at a concentration range of 0.5–1,000 ng/ml.

6  An aliquot of plasma (50 µl) was precipitated with acetonitrile (MeCN) (300 µl) containing an internal standard.

7  Samples were vortex-mixed (1 min) and centrifuged (180 rcf for 10 min) to afford supernatant (250 µl), which was transferred to a 96-well plate.

8  Supernatants were evaporated under $N_2$ at 37°C and reconstituted in 25% MeCN in $H_2O$ (100 µl).

Samples were analyzed by an LC-MS/MS comprising an AB Sciex API 4000 tandem quadrupole mass spectrometer (AB Sciex Inc., Ontario, Canada) with a TurboIonSpray probe (AB Sciex Inc., Ontario, Canada), tertiary Shimadzu LC20AD pumps (Shimadzu Scientific Instruments, Columbia, MD), and a CTC PAL autosampler (Leap Technologies, Carrboro, NC). Instrument settings and potentials were adjusted to provide optimal data. All raw data were processed using Analyst Software version 1.4.2 (AB Sciex Inc., Ontario, Canada).

PF-670462 and its internal standard within sample aliquots (5 µl) were eluted at 0.5 ml/min on a Synergi Max RP analytical column (4 µ, 80 Å, 2 × 30 mm; Phenomenex, Torrance, CA) using 10 mM ammonium formate in 0.1% $HCO_2H$ (solvent A) and MeCN (solvent B) via the following gradient: 0 to 0.75 min, 5% solvent B; 0.75 to 1.25 min, 5 to 90% B; 1.25 to 2.00 min, hold at 90% B; and 2.00 to 2.5 return to 5% B and hold until 3.00 min. Mass spectral data were collected in positive electrospray ionization mode using multiple-reaction monitoring (MRM) following the $m/z$ 338.2→256.2 fragmentation for PF-670462.

### Plasma protein binding

The plasma-free fraction of PF-670462 was determined in monkey plasma by equilibrium dialysis using a previously reported method (Kalvass & Maurer, 2002).

1  Aliquots (200 µl, $n = 6$) of freshly harvested monkey plasma containing PF-670462 (1 µM) were loaded into a 96-well equilibrium dialysis apparatus (HTDialysis, Gales Ferry, CT) containing preconditioned Spectra-Por2 membranes (Spectrum Laboratories Inc., Rancho Dominguez, CA; molecular weight cutoff 12,000–14,000).

2  Plasma was dialyzed against an equal volume of phosphate-buffered saline for 6 h at 37°C.

3  After incubation, 150 µl of plasma sample and 150 µl of buffer sample were removed from the apparatus and matrix-matched for bioanalysis by addition of an equal volume of opposing matrix to yield identical sample composition between buffer and non-buffer samples.

Protein binding samples were analyzed by HPLC/MS/MS using a non-validated bioanalytical assay.

4  An aliquot of matrix-adjusted sample (50 µl) was precipitated with acetonitrile (MeCN) (200 µl) containing an internal standard.

5  Samples were vortex-mixed (2 min) and centrifuged (1,811 rcf for 7 min) to afford supernatant (150 µl), which was transferred to a 96-well plate.

Samples were analyzed by an LC-MS/MS comprising an AB Sciex API 4000 tandem quadrupole mass spectrometer (AB Sciex Inc., Ontario, Canada) with a TurboIonSpray probe (AB Sciex Inc., Ontario, Canada), tertiary Shimadzu LC20AD pumps (Shimadzu Scientific Instruments, Columbia, MD), and a CTC PAL autosampler (Leap Technologies, Carrboro, NC). Instrument settings and potentials were adjusted to provide optimal data. All raw data were processed using Analyst Software version 1.4.1 (AB Sciex Inc., Ontario, Canada).

PF-670462 and its internal standard within sample aliquots (10 µl) were eluted at 0.4 ml/min on a Luna C18 analytical column (5 µ, 80 Å, 2.1 × 30 mm; Phenomenex, Torrance, CA) using 50% 20 mM ammonium acetate in 0.1% isopropyl alcohol, and 50% acetonitrile (solvent A) and MeCN (solvent B) via the following gradient: 0–0.5 min, 10% solvent B; 0.5–1.5 min, 10–90% B; 1.5–2.20 min, hold at 90% B; and 2.21 return to 10% B and hold until 3.00 min. Mass spectral data were collected in positive electrospray ionization mode using MRM following the $m/z$ 338.2→256.2 fragmentation for PF-670462.

The plasma-free fraction was calculated from the ratio of concentrations determined from the plasma and buffer samples.

### Estimation of free brain concentration

Free PF-670462 brain concentration was estimated from plasma by first correcting for the fraction of unbound PF-670462 in plasma (fu = 0.15) and subsequently correcting for the free brain and free plasma concentration ratio (Cbu/Cpu) of 1, an approach commensurate with unrestricted equilibrium between the central and plasma compartments as predicted by physiologically based PK modeling (Trapa et al, 2016).

### Modeling studies
#### Mathematical model description

We adopted our previous systems chronopharmacology model (274 variables and 86 parameters), which successfully investigated the effect of PF-670462 in mice (Kim et al, 2013). This model was developed by extending the Kim–Forger model (Kim & Forger, 2012), a detailed mathematical model of the intracellular mammalian circadian clock, which describes the reactions among core clock molecules (e.g., binding, phosphorylation, subcellular translocation, transcription, and translation) using ordinary differential equations based on mass action kinetics (181 variables and 75 parameters). To develop the systems chronopharmacology model for NHP (Fig 2A), the original mouse model (Kim et al, 2013) was modified by newly estimating the pharmacokinetic (PK) parameters (Fig 2Ai) and incorporating the gating and adaptation into the light module of the model (Fig 2Aii).

#### Modification of the PK parameters

The six parameters describing the PK properties of PF-670462 (e.g., transfer rate between plasma and brain tissue) were modified due to the difference in free PF-670462 exposure in brain tissues (Fig 1B) and its effect in DD (Fig 1D) between NHPs and mice. Specifically, the parameters were fitted to the disposition profiles of PF-670 and its DD dosing effect in NHPs (see Fig EV1A–C and Dataset EV1 for details).

#### Incorporation of gating and adaptation for light into the model

To incorporate the gating into the model, we used the function $g$, which determines the photosensitivity of the circadian clock at each CT (Fig EV1F). To connect the photo-insensitive and photosensitive

zones of $g$, a piecewise polynomial interpolation was used (see Code EV1). To incorporate the adaptation into the model, we used the Hill function, which expresses light duration-dependent reduction in photosensitivity (Fig EV1H).

The values of six parameters determining the functions of gating and adaptation (Dataset EV2) were estimated with the simulated annealing (SA) method (Gonzalez *et al*, 2007) and post-filtering in two steps. In the first round, we found 991 parameter sets with which the model accurately simulates the phase delay of NHPs induced by the 3-day LD dosing and the magnitude of human PRC to a 6.7-h light pulse (Fig EV2A) using SA method with the cost function:

$$\sqrt[2]{\left(\frac{\min(m - \tilde{m} + 1, 0)}{\tilde{m} - 1}\right)^2 + \left(\frac{\max(m - \tilde{m} - 1, 0)}{\tilde{m} + 1}\right)^2 + \left(\frac{\min(M - \tilde{M} + 1, 0)}{\tilde{M} - 1}\right)^2 + \left(\frac{\max(M - \tilde{M} - 1, 0)}{\tilde{M} + 1}\right)^2} + f_1 + \sqrt{\sum_{j=1}^{3}\left(\frac{x_j - \tilde{x}_j}{\tilde{x}_j}\right)^2}.$$

where $f_1(x) = \begin{cases} 10^3 \text{ If } \max(|m|, |M|) \geq 6 \\ 0 \text{ If } \max(|m|, |M|) < 6 \end{cases}$.

$m$ and $M$ are the min and max of the simulated PRC to a 6.7-h light pulse, respectively. $\tilde{m} = -3.26$ and $\tilde{M} = 2.63$ h are the min and max of the human PRC to a 6.7-h light pulse, respectively (Khalsa *et al*, 2003). $x_j$ and $\tilde{x}_j$ are the simulated phase delay and experimentally measured phase delay (Fig 1D) on day $j$ of 3-day LD dosing, respectively.

In the second round, using post-filtering, we filtered in the final 10 parameter sets (Dataset EV2) with which the model accurately simulates the type 1 PRC to a 12-h light pulse and the human PRC to a 6.7-h light pulse and to 3-cycle 5-h light pulses (Fig EV2).

To estimate the input CT for $g$ even when the circadian phase is perturbed, we constructed the function $p$ which determines the CT from an internal pace marker: the phase angle of the limit cycle of two clock variables, *revng* and *revnp* in the model (Fig EV1G). We first interpolated the phase angles of the limit cycle of *revng* and *revnp* to the CTs. Then, we composed the interpolation function with the function, $\tan^{-1}$ (*revng*(t), *revnp*(t)), which estimates the phase angle of the limit cycle from the concentrations of *revng* and *revnp* at $t$. As $g \cdot p$ is the composite of the interpolation functions, which do not have explicit form, including $g \cdot p$ into the model increases the computational cost of the simulation. Thus, the approximated $g \cdot p$ using Fourier series was used, which accurately determines the phase-dependent photosensitivity even when the circadian phase is altered by PF-670462 or light.

### Development of the ASPD models

To develop the ASPD models (Dataset EV3), we investigated the modification of which parameter allows for advancing the phase by ~4 h from the WT model (Fig 4A), reflecting the advanced circadian phase of ASPD patients (Jones *et al*, 1999). In this process, to reflect the advanced circadian phase, the phase of gating function g is also advanced by 4 h.

### Model assumptions

1  The model of the intracellular mammalian circadian clock, Kim–Forger model, which was developed to accurately simulate the mouse SCN (Kim & Forger, 2012), was used for the new NHP

model as clock gene expression profiles in the SCN of cynomolgus monkeys have not been measured. Nevertheless, due to the new light module, the NHP model simulates more advanced clock gene expression compared with the original mouse model under LD, which is consistent with the advanced clock gene expression seen in the SCN of baboons compared with that of mice under LD (Mure *et al*, 2018).

2  The pharmacodynamic parameters, describing the intracellular action of PF-670462 (e.g., binding of PF670 with CK1$\delta$/$\epsilon$), of the original systems chronopharmacology model (Kim *et al*, 2013), were kept as they are expected to be similar between NHPs and mice.

3  The reduced photosensitivity due to adaptation for light during daytime is assumed to be completely recovered after nighttime if it is long enough ($\geq 8$ h).

4  To simulate the phase shift of activity onsets with the model, we used the phase shift of the simulated BMAL1 gene expression profile as the phase of BMAL1 gene expression profiles and activity onset is highly correlated (Kiessling *et al*, 2010). However, as the phases of all clock gene expression profiles are tightly interlocked in the model, the result changes little even if the phase of other clock gene expression profiles is used.

### Simulation

All the simulations and parameter searches were performed using MATHEMATICA 11.0 (Wolfram Research, Champaign, IL) with a computer cluster composed of seven machines where each machine is equipped with two Intel Xeon SP-6148 CPUs (2.4 GHz, 20C), 192GB RAM, and the operating system CentOS 7.4 64bit.

### Statistical analysis

For this study, one-way ANOVA, two-way ANOVA, and Pearson's correlation test were used to determine statistical significance. Outliers were detected by Grubbs' test. They are each detailed along with the relevant parameters in the main text, figures, and figure legends. A *P*-value of < 0.05 was considered statistically significant. One-way ANOVA, two-way ANOVA, and Pearson's correlation test were performed using MATHEMATICA 11.0. Grubbs' test was performed using GraphPad Quickcalcs software (GraphPad Software Inc, La Jolla, CA, USA, available at http://www.graphpad.com/quickcalcs/index.cfm).

## Data availability

The MATHEMATICA codes used in this study are available in Code EV1 and the following database: https://github.com/daewookkim/Non-human-primate-circadian-clock-model-including-CK1-inhibitor.

**Expanded View** for this article is available online.

## Acknowledgements

We thank Boseung Choi, Daniel B. Forger, and David M. Virshup for valuable comments and Life Science Editors for editorial assistance. This work was supported by a Pfizer grant to Korea Advanced Institute of Science and Technology (G01160179), the Human Frontiers Science Program Organization (RGY0063/2017), and a National Research Foundation of Korea Grant funded by the Korean Government (NRF-2016 RICIB 3008468 and NRF-2017-Fostering Core Leaders of the Future Basic Science Program/Global Ph.D. Fellowship Program).

## Author contributions

CC, GJD, TW, and JKK designed the study. GJD and ACD performed the experiments. DWK and JKK performed the computational modeling and mathematical analysis. DWK, GJD, XC, ACD, and JKK analyzed the experimental data. CC, FG, and JKK designed the translation strategy. DWK and JKK wrote the manuscript, and all authors contributed to reviewing the manuscript.

## Conflict of interest

The authors declare that they have no conflict of interest.

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
