## [Review Process File · Molecular Systems Biology]

Systems approach reveals photosensitivity and PER2 level as determinants of clock-modulator efficacy

Authors: Dae Wook Kim, Cheng Chang, Xian Chen, Angela Doran, Francois Gaudreault, Travis Wager, George J. DeMarco, Jae Kyoung Kim.

Review timeline:	Submission date:	23 rd January 2019
	Editorial Decision:	4 th March 2019
	Revision received:	16 th April 2019
	Editorial Decision:	29 th May 2019
	Revision received:	30 th May 2019
	Accepted:	3 rd June 2019

Editor: Jingyi Hou

Transaction Report:

1st Editorial Decision

4th March 2019

Thank you for submitting your work to Molecular Systems Biology. We have now heard back from the three reviewers who agreed to evaluate your study. As you will see below, the reviewers think that the presented findings seem interesting. They raise however a series of concerns, which we would ask you to address in a revision.

I think that the recommendations provided by the reviewers are clear so there is no need to repeat the points listed below. All issues raised by the reviewers need to be convincingly addressed. As you may already know, our editorial policy allows in principle a single round of major revision so it is essential to provide responses to the reviewers' comments that are as complete as possible. Please feel free to contact me in case you would like to discuss in further detail any of the issues raised by the reviewers.

In line with the comments of all the reviewers, we would ask you to provide the full details of your model.

REFeree REPORTS

Reviewer #1:

This is an interesting and potentially useful translational study that uses a systems approach to validate models concerning CK1delta/epsilon inhibition in non-human primates. The authors show that there are important species differences between the commonly used pre-clinical nocturnal mouse model and primates. This is likely to have important consequences for developing and applying novel chronotherapeutic approaches in the field (i.e. in clinical trials and, eventually, in the clinical arena).

In general, the manuscript is presented nicely and the data sound. I do, however, make a few suggestions below to help improve the paper.

Major points

1. The authors' primary conclusion is that interspecies differences of response to light (and CKId/e inhibition) is because of differences in PER2 protein levels. Their conclusions would be significantly strengthened if they could show that absolute difference in this in the different species. As far as I can see, there is only modelling data suggesting this on the basis of shifts, rather than molecular quantifications to validate this. This may in turn augment the proposed model and/or refine parameters.
2. The authors have made their models using Mathematica and say that the code is available. The authors should deposit the models and instructions on Github or similar, along with full details of parameter search and scripts for implementation on their systems (150 nodes) so that the models can be easily and independently validated by others in the future.

Minor points

1. The text is generally quite long and a bit labored in places. It should be made more concise and easy to read with additional editing.
2. The authors refer to "PF-670" throughout the manuscript. They should, however, refer to it by its full designation (PF-670462) in the methods section to avoid any confusion for readers who are primarily interested in these details.

Reviewer #2:

The introduction well summarizes the present situation of chronotherapy and points out the importance of a quantitative model that predicts circadian phase responses to light or pharmacological perturbations. To create such a model, the authors updated their previous model describing the detailed molecular events of mammalian circadian clocks as well as the effect of CKI kinase inhibitors (CKIi). By incorporating the gating and adaptation processes of the light signals, the extended model recaptured the experimental observation of macaques' phase-shift response upon CKIi administration-unlike the case in mice, the drug effect is reduced in LD condition in macaques. The model reasoned that this difference is caused by different photosensitivity resulting in varied PRCs. The model also predicts that the effect of CKIi depends on the level of PER protein at the timing of drug administration. Because the impact of CKIi on the phase shift changes depending on the administration timing, the model is used to demonstrate the iterative approach for optimizing the appropriate drug dose and administration timing. Overall, the authors well summarized the complex but precise modeling study. The role of light and PER expression on the phase-modulation effect of CKIi will be informative for the circadian experimental biologist. Furthermore, the model itself is highly valuable for circadian medicine. A few points listed below should be clarified before publication.

Major comments:

- 1) Describe the full detail of the model used in this study. It is important not only for the manuscript's integrity but also for the future use of the model to optimize the dosing protocol by other researchers. The core models were described in two previous papers (Kim 2013, 2012), yet the present study may have optimized some parameters. Please consider to describe the full equations and a parameter list in the Appendix. The method/equations to incorporate the function of g and p should be explicitly written.
- 2) On the results shown in figure 1E-F, the authors mainly focused on the different lighting condition. However, the timings of CKIi exposure were different between LD and DD conditions (ZT11 and CT14). Explain the rationale to exclude the possibility that the different drug exposure

timing is not the reason for the different phase shift patterns between mice and macaques.

3) The relationship between "photosensitivity" and "nocturnal/diurnal" may be confusing in several contexts. The model modulates the "photosensitivity" that could be represented by the different curves of PRC, but this is not necessarily accounting for the nocturnal/diurnal difference. For example, mice can be diurnal in some conditions. Is there evidence indicating that nocturnal/diurnal differences account the different photosensitivity represented as the current model?

4) I am not entirely convinced by the authors' conclusion stating that "we found that such variability in the CKIi effect is mainly due to altered PER2 abundances" (page 16), because the analysis shown in figure 4 is limited to the correlation only between phase shift and PER2 level. By using the model, it would be reasonable to ask whether the PER2 level is the only factor that satisfactorily explains the different phase shift (by calculating the correlation between phase shift and many other parameters). Conversely, it would be possible to tune the parameters to change the expression level of PER2 but not change the free-running period and confirm the correlation against phase shift upon CKIi challenge. It would also be important to show that the current model recapitulates the correlation shown in figure 4E.

Related to this, please explain the reason why the phase shift did not change between ZT4 dosing and ZT11 dosing (figure 2) where the level of PER2 should change in wildtype NHPs.

Minor comments:

5) Abstract: "a systems pharmacology model" does not summarize any model structures. Please briefly mention what types of model (e.g., a model describing the detailed molecular reactions, etc.) was used in this study. Note that I am not requesting to remove the word "systems pharmacology model."

6) Figure 4C and 4D: please indicate the points of each mutation (not just by showing the name of mutants). Also, several overwrapped characters (name of CRY mutants) should be amended.

7) The adaptive chronotherapeutic approach shown in figure 5 works well. However, one may naively think that the iterative adaptation works in any case without the help of model prediction. Please consider providing more rationale and discussion about how the model prediction provides the basis of the adaptive chronotherapeutic protocol introduced in this study.

8) This is a very minor comment for figure 5A; what do the sun and snow marks at the upper-left of humans represent?

Reviewer #3:

The authors address an interesting and relevant interdisciplinary topic - the treatment of advanced sleep phase disorder (ASPD) via CKI inhibitor. The authors compare novel primate data with previously published mouse data (Fig.1) and adapt existing mathematical models accordingly (Fig.2). It is suggested that differential gating mechanisms modulate the effects of CKI inhibitors. Consequently, treatment of ASPD cannot be based only on mouse experiments. In addition, gating models and some data indicate that PER2 levels influence the phase shifts via inhibitor (Fig.4). The systemic understanding of treatment allows predictions of relatively simple strategies to adapt dosing regimen (Fig.5).

My major concerns refer to the limits of quantitative modeling. In my eyes, there is not a single quantitative model of a eukaryotic clock since multiple transcription factors (see Ueda reviews), epigenetic regulations (Sassone-Corsi, Takahashi), huge protein complexes of unknown stoichiometry (Weitz), multiple phosphorylations (Virshup, Kramer) etc. do not allow detailed and precise modeling. The last author, however, is well aware of the limitations of modeling and employs ensembles of reasonable models (Methods, Appendix). Consequently, the main results (phase shifts, differences between night- and day-active animals, role of PER2) seem to be independent of modeling details. Thus the proposed treatment strategy seems quite useful despite known limitations of quantitative modeling.

Specific comments:

1. The known phase shift due to electrical illumination should not be called shift work.
2. Page 7 "... these processes are slowed down" The connection of to PER2 stability, nuclear import and export and periods are quite complex (see, e.g., papers by Vanselow and Relogio). Period shortening and lengthening can be explained by similar mechanisms.
3. Recently, expression profiles in baboon data have been published providing despite limited sampling some reasonable phases and amplitudes of core clock genes. Are the differences to mouse data (e.g. from Hogenesch) connected to the discussed differential phases responses?
4. Some comments on the number of new model parameters, fitting procedures, limitations of models might be added to the main text. Without reading previous papers of the last author and of the Appendices it is difficult to develop an understanding of the underlying models (e.g. the meaning of models N and W are unclear from the main text).
5. It might be stressed more clearly how data have been used to constrain models (training sets) and what data are consistent with experiments without explicit fitting.
6. It should be discussed how the acute circadian phase can be estimated to adapt control. There are traditional techniques (activity, melatonin onset, body temperature (R. Wever book)) and newly developed markers (Ueda, Dallmann, Kramer).

We thank the reviewers for their positive feedback on our manuscript and constructive comments and suggestions. In response, we have heavily revised the manuscript. Below, we give our detailed responses to the reviewers' comments and describe the changes in the manuscript. The reviewers' comments appear in black, and our responses in blue.

Reviewer #1:

This is an interesting and potentially useful translational study that uses a systems approach to validate models concerning CK1delta/epsilon inhibition in non-human primates. The authors show that there are important species differences between the commonly used pre-clinical nocturnal mouse model and primates. This is likely to have important consequences for developing and applying novel chronotherapeutic approaches in the field (i.e. in clinical trials and, eventually, in the clinical arena).

In general, the manuscript is presented nicely and the data sound. I do, however, make a few suggestions below to help improve the paper.

Major points

1. The authors' primary conclusion is that interspecies differences of response to light (and CK1delta/epsilon inhibition) is because of differences in PER2 protein levels. Their conclusions would be significantly strengthened if they could show that absolute difference in this in the different species. As far as I can see, there is only modelling data suggesting this on the basis of shifts, rather than molecular quantifications to validate this. This may in turn augment the proposed model and/or refine parameters.

We apologize for the confusion. We investigated the relationship between difference in PER2 abundance and the variation in the effect of the CK1delta/epsilon inhibitor at the *intraspecies* level, but not at the *interspecies* level. Specifically, we found that PER2 abundance is a key determinant of the CK1delta/epsilon inhibitor effect by identifying the positive correlation between PER2 abundance of various ASPD models and their response to CK1delta/epsilon inhibition (Fig 4C-E and EV5B-F). Furthermore, this was supported by the experimental data: the strong positive correlation between PER2 abundance when dosing occurs (Amir et al, 2004) and the phase delay induced by CK1delta/epsilon inhibition (Badura et al, 2007) (Fig 4E). However, we did not investigate whether different PER2 protein abundance between mice and non-human primates (NHPs) causes their different response to CK1delta/epsilon inhibition and light. In fact, although our mouse model (Kim et al, CPT:PSP, 2013) and NHP model (Fig 2A) have nearly the same PER2 abundance (Fig R1), they recapitulated the experimentally measured interspecies difference in the effect of the CK1delta/epsilon inhibitor (Fig 1E-G and 3A) (Kim et al, CPT:PSP, 2013). Specifically, the stronger DD dosing effect in NHPs than in mice (Fig 1E) was reproduced by the models because the different pharmacokinetic parameters (Fig 2Ai and EV1A-C) were used reflecting the higher drug exposure in NHPs than in mice (Fig 1B). Furthermore, the stronger light-induced attenuation of the drug effect in NHPs than in mice (Fig 1E-G and 3A) was captured by adopting the different light modules (Fig 2Aii and EV1F-H) so that the NHP model can simulate a larger magnitude of advance zone of light phase response curve (PRC) than the mouse model (Fig 3B) (Kim et al, CPT:PSP, 2013). The light PRC of NHPs can have a larger

magnitude of advance zone than that of mice for various reasons (e.g. differences in CREB phosphorylation rate, pCREB binding rate to Per promoter CRE site or melanopsin sensitivity) (Ginty et al, Science, 1993; Pulivarthy et al, PNAS, 2007; Rollag et al, J Biol Rhythms, 2003). In particular, as the reviewer suggested, PER2 abundance also could influence the shape of light PRC because, as PER2 abundance increases, the magnitude of light-induced phase advance and delay often decreases (Pulivarthy et al, PNAS, 2007). However, identifying whether the difference in PER2 abundance leads to the different shape of light PRC between mice and NHPs is difficult because measuring the absolute PER2 abundance in SCN is extremely challenging (see below). Furthermore, note that although light PRC has been measured throughout the nearly entire history of chronobiology, the detailed molecular mechanisms regulating the magnitude of either advance or delay zone of the light PRC has been poorly understood (Golombek & Rosenstein, Physiol Rev, 2010; Ripperger & Brown, New York: Springer (2010), pp. 37-78).

Figure R1. PER2 protein abundances simulated by the mouse model (Kim et al, CPT:PSP, 2013) and the NHP model (Fig 2A) are nearly the same. The abundances were normalized by the maximum abundance simulated by the mouse model in LD 12:12. The white and black rectangles indicate the times of light going on and off.

As PER2 abundance is a determinant of the effect of the CK1 δ/ϵ inhibitor at the *intraspecies* level (Fig 4C-E and EV5B-F), we agree with the reviewer that the *interspecies* difference in PER2 abundance could also lead to the *interspecies* difference in response to CK1 δ/ϵ inhibition. To investigate this, the quantification of the *absolute* PER2 protein abundance in the SCN of mice and primates is required. Previously, PER2 protein abundance in the mouse SCN has been measured only at the *relative* level using immunocytochemistry (Reppert and Weaver, Annu Rev Physiol, 2001). It is only lately that the *absolute* PER2 abundance in the mouse liver has been quantified using mass spectrometry-based proteomics, thanks to the enormous effort of the Hiroki Ueda group (Narumi et al, PNAS, 2016). However, applying mass spectrometry to the SCN whose size is much smaller than that of liver tissue, appears to be much more challenging. Due to the technical challenges, neither PER2 protein abundance nor its gene expression has been measured in the SCN of primates. Only gene expression of Per2 in the peripheral tissues of primates has been recently measured as pointed out by reviewer 3 (Mure et al, Science, 2018). Thus, although investigating the interspecies difference in PER2 protein abundance is important, it seems difficult with current experimental techniques. However, we did feel that this comment was important, so we have discussed it as future work:

- P18 L380: “We found that such variability in the CK1i effect is mainly due to altered PER2 abundances (Fig 4C-E and EV5B-F). *While the interspecies difference in PER2 abundance has not been investigated, it may contribute to the interspecies variability in the CK1i effect (Fig*

1E-G and 3A). Furthermore, the interspecies difference in the phase of PER2 rhythms (Millius & Ueda, 2018; Mure et al, 2018; Vosko et al, 2009; Zhang et al, 2014), which leads to the interspecies difference in PER2 abundance when dosing occurs, could also be the source of the interspecies variability in the CK1i effect. It would be interesting to investigate such a relationship between the effect of CK1i and PER2 abundance, and extend this to other clock-modulating drugs and their target molecules (e.g. KL001 targeting CRY (Hirota et al, 2012)).”

2. The authors have made their models using Mathematica and say that the code is available. The authors should deposit the models and instructions on Github or similar, along with full details of parameter search and scripts for implementation on their systems (150 nodes) so that the models can be easily and independently validated by others in the future.

The code EV1 containing the codes of the NHP model (Fig 2A) and the parameter search and their instructions has been added to the manuscript. Furthermore, we have deposited them on Github, which will be made public when the manuscript is accepted. We have also added Appendix Equation S1, Table EV1 and 2 and Dataset EV1-3 to describe the model equations, variables and parameters. Furthermore, we have revised the Materials and Methods to describe the model more clearly (see below, comment 4 of reviewer 3 for details).

Minor points

1. The text is generally quite long and a bit labored in places. It should be made more concise and easy to read with additional editing.

We thank the reviewer for pointing this out. We have revised the text more readable. However, unfortunately, the overall length of the manuscript has not been significantly reduced because new materials have been added in the revision. Furthermore, it was difficult to reduce the details of the manuscript (both experiments and modeling) considering the diverse backgrounds of potential readers for this manuscript (biologists, mathematicians, medical doctors...).

2. The authors refer to "PF-670" throughout the manuscript. They should, however, refer to it by its full designation (PF-670462) in the methods section to avoid any confusion for readers who are primarily interested in these details.

We have now referred to PF-670 by its full designation in the Materials and Methods.

Reviewer #2:

The introduction well summarizes the present situation of chronotherapy and points out the importance of a quantitative model that predicts circadian phase responses to light or pharmacological perturbations. To create such a model, the authors updated their previous model describing the detailed molecular events of mammalian circadian clocks as well as the effect of CKI kinase inhibitors (CKIi). By incorporating the gating and adaptation processes of the light signals, the extended model recaptured the experimental observation of macaques'

phase-shift response upon CKIi administration-unlike the case in mice, the drug effect is reduced in LD condition in macaques. The model reasoned that this difference is caused by different photosensitivity resulting in varied PRCs. The model also predicts that the effect of CKIi depends on the level of PER protein at the timing of drug administration. Because the impact of CKIi on the phase shift changes depending on the administration timing, the model is used to demonstrate the iterative approach for optimizing the appropriate drug dose and administration timing. Overall, the authors well summarized the complex but precise modeling study. The role of light and PER expression on the phase-modulation effect of CKIi will be informative for the circadian experimental biologist. Furthermore, the model itself is highly valuable for circadian medicine. A few points listed below should be clarified before publication.

Major comments:

1) Describe the full detail of the model used in this study. It is important not only for the manuscript's integrity but also for the future use of the model to optimize the dosing protocol by other researchers. The core models were described in two previous papers (Kim 2013, 2012), yet the present study may have optimized some parameters. Please consider to describe the full equations and a parameter list in the Appendix. The method/equations to incorporate the function of g and p should be explicitly written.

We thank the reviewer for this comment. We have now further described the model by adding the full model equations, variables and parameters (see below, comment 4 of reviewer 3). Unfortunately, as the function of g and p was constructed using *Interpolation*, which is a built-in function in *Mathematica*, it cannot be explicitly written. Thus, we have added code EV1 containing the code of the NHP model (Fig 2A) and its instructions to the manuscript, which clearly describe the construction procedure of the function g and p . Furthermore, to illustrate the function g and p , we have added a new figure (Fig. EV1G):

Figure EV1. The original model (Kim et al, 2013) with modified pharmacokinetic parameters accurately captures DD dosing but not LD dosing in NHPs.

F To accurately capture the effect of LD dosing in NHPs, gating for light, which is denoted by function g in Materials and Methods and Appendix Equation S1, is incorporated into the

model (Fig 2Aii). The shape of the gating is determined by four parameters: X_1 determines the circadian time when the gating becomes weaker and thus the photosensitivity increases. X_2 describes the range of high photosensitivity zones. Y_1 describes the photosensitivity of the circadian clock when it is fully inhibited, which is assumed to be constant for simplicity. Y_2 describes the maximum photosensitivity of the circadian clock. To connect these photosensitive and photosensitive zones continuously, a piecewise polynomial interpolation is used (see code EV1). Note that the gating depends on the CT.

G *To estimate the input CT for gating (F) even when the circadian phase is altered by a stimulus (i.e. light and PF-670), we constructed the function p , which estimates the CT from the phase angle of limit cycle of two clock variables, $revng$ and $revnp$ (Table EV1). (i) When the circadian clock is entrained by external light (i.e. LD 12:12), the CT can be simply approximated by ZT (e.g. $CT_{12} \approx ZT_{12}$; blue circle). However, if the circadian phase is delayed by PF-670, ZT, which corresponds to the same CT (e.g. CT_{12}) changes dramatically (e.g. $ZT_{14, 16, 19}$ and 23 ; red circles). (ii) On the other hand, the phase angles of limit cycle of $revng$ and $revnp$, which corresponds to the same CT, change little (e.g. blue and red circles corresponding to CT_{12} in (i)). (iii) Based on this feature, we constructed the function p , which is the function of the phase angle of the limit cycle (i.e. $\tan^{-1}(revng(t), revnp(t))$) for CT when the model is entrained to LD 12:12 (gray dashed line). This allows the model to accurately predict the CT from the phase angle even when the circadian phase is altered by PF-670 (orange range). The orange range represents the mean \pm SD of the predicted CTs using the phase angle when a single daily 30 mpk dosing is given at ZT_{14} for 20 days. Note that p is accurate up to a considerably high dose (~ 80 mpk) as the limit cycle is stable.*

H *The adaptation for light is incorporated into the model (Fig 2Aii). The shape of adaptation is described with a Hill function with two parameters: Z_1 and Z_2 determine the light duration which reduces the photosensitivity by 50% and how sharply the photosensitivity decreases, respectively.*

2) On the results shown in figure 1E-F, the authors mainly focused on the different lighting condition. However, the timings of CK1i exposure were different between LD and DD conditions (ZT_{11} and CT_{14}). Explain the rationale to exclude the possibility that the different drug exposure timing is not the reason for the different phase shift patterns between mice and macaques.

Thanks for pointing this out. In this manuscript, we observed that individual non-human primates (NHPs), which were treated with 10 mpk PF-670462 (PF-670), the CK1 δ/ϵ inhibitor, at various dosing times in DD for 3 days, showed a larger phase delay (4.4 h; CT_4 , 5.8 h; CT_6 , 5.3 h; CT_8 and 4.6 h; CT_{14}) than mice dosed with 32 mpk PF-670 for 3 days in DD (3.9 h; CT_{11}) (Fig 1E and EV1C and EV3). However, the phase delay of NHPs induced by the 3-day 10 mpk LD dosing at ZT_{11} (1.7 h) was not larger than that of mice induced by the 3-day 32 mpk LD dosing at ZT_{11} (2.2 h) (Fig 1F and EV3). From these data, we concluded that light attenuates the PF-670-induced phase delay more strongly in NHPs than in mice. This conclusion could be wrong if the phase delay of NHPs induced by the 3-day 10 mpk DD dosing at CT_{11} is not larger than that of mice induced by 3-day 32mpk DD dosing at CT_{11} . However,

this is unlikely due to the following reasons.

Despite the lower dose level in NHPs (10 mpk) than in mice (32 mpk), the drug exposure in brain tissues is ~7-fold higher in NHPs ($AUC=3.6 \mu\text{M} \cdot \text{h}$) than in mice ($AUC=0.5 \mu\text{M} \cdot \text{h}$) (Fig 1B). Thus, all NHPs, which were treated with 10 mpk PF-670 at various dosing times (CT4, CT6, CT8 and CT14) in DD, showed a larger phase delay than mice dosed with 32 mpk PF-670 at CT11 in DD, although dosing at CT11 leads to a larger phase delay than dosing at other CT (Badura et al, J. Pharmacol. Exp, 2007) (Fig 1E and EV1C and EV3). Similarly, due to the much higher drug exposure in brain tissue of NHPs than mice (Fig 1B), NHPs dosed with 10 mpk PF-670 at CT11 are expected to show a larger phase delay than mice dosed with 32 mpk at CT11 in DD. This is supported by simulation of the NHP model (blue line; Fig EV3), which accurately reproduces the effect of DD dosing in NHPs at various dosing times (Fig EV1C and EV3). Note that the dosing of PF670 at CT11 leads to a similar or larger phase delay than dosing at other times in NHP (Fig EV3) consistent with the experimentally measured PRC (Badura et al, J. Pharmacol. Exp, 2007) (i.e. the phase delay of NHPs dosed at $CT11 \geq$ the phase delay of NHPs dosed at other CTs $>$ the phase delay of mice dosed at CT11). As these points were not described in the manuscript, we have revised the manuscript as follows:

- P8 L148: “To analyze the effect of PF-670 on the circadian phase in diurnal NHPs, and compare it with nocturnal mice, we first compared the free PF-670 brain concentrations across species (see Materials and Methods for details). Despite the lower dose level in NHPs (10 mg/kg (mpk)) than used in our previous study in mice (32 mpk), the drug exposure in NHPs ($AUC=3.6 \mu\text{M} \cdot \text{h}$) was much higher than in mice ($AUC=0.5 \mu\text{M} \cdot \text{h}$) (Fig 1B) (Kim et al, 2013). Due to the higher drug exposure in NHPs, we hypothesized that PF-670 induces a larger phase delay of activity onset in NHPs than in mice. To investigate this, we compared the phase delays of NHPs induced by 3-day 10 mpk dosing in a dark-dark cycle (DD) (Fig 1C) with the phase delays of mice dosed with 32 mpk PF-670 for 3 days in DD (Kim et al, 2013). Indeed, NHPs showed a significantly larger phase delay (5.2 h) compared to mice (3.8 h) ($P=0.03$; Fig 1E) (see Materials and Methods for details of phase delay measurement). *This larger phase delay in NHPs than in mice might be due to the different dosing times for NHPs (e.g. CT14) than for mice (CT11) as the effect of PF-670 changes upon dosing time (Badura et al, 2007). However, the phase delay of NHP induced by dosing at CT11 is also likely to be larger than that of mice because the dosing at CT11 is expected to yield a nearly maximal phase delay (Badura et al, 2007) and the drug exposure is much higher in NHPs than in mice (Fig 1B) (see Fig EV3 after reading below two sections for details).”*

Figure EV3. The phase delays induced by 3-day PF-670 dosing in NHPs and mice under DD and LD. NHPs and mice were treated with PF-670 for 3 days (Fig 1E and F and 2F and EV1C). Due to the higher drug exposure in NHPs than in mice (Fig 1B), all NHPs, which were treated with 10 mpk PF-670 at various dosing times (CT4, 6, 8, 14) in DD, show a larger phase delay (blue circles) than mice dosed with 32 mpk PF-670 at CT11 in DD (blue square). Thus, the phase delay of NHPs induced by 10 mpk PF-670 dosing at CT11 is expected to be larger than that of mice dosed with 32 mpk PF-670 at CT11 under DD, which is supported by the model simulation (blue line). Due to the strong attenuation of the PF-670 effect by light in NHPs, the change of drug effect upon dosing time is much smaller in LD (from 0.6h to 2.1h; red arrow) than in DD (from 0.8h to 6.1h; blue arrow). Dosing time is denoted by the x-axis. The line and colored range represent the mean \pm SD of the simulated phase delays of NHP models with the 10 pairs of gating and adaptation (Fig 2Aii and EV2F).

3) The relationship between "photosensitivity" and "nocturnal/diurnal" may be confusing in several contexts. The model modulates the "photosensitivity" that could be represented by the different curves of PRC, but this is not necessarily accounting for the nocturnal/diurnal difference. For example, mice can be diurnal in some conditions. Is there evidence indicating that nocturnal/diurnal differences account the different photosensitivity represented as the current model?

We completely agree with the reviewer that the "photosensitivity", which determines the ratio of maximum magnitudes between the advance (A) zone and the delay (D) zone (A/D ratio) of PRC, does not necessarily account for the nocturnal/diurnal difference. As the reviewer pointed out, even nocturnal animals could be active in daytime for some circumstances (e.g. food availability, social pressures, seasonal cues and physiological or molecular manipulation) (Mrosovsky, *Chronobiol Int*, 2003, Mrosovsky & Hatter, *J Comp Physiol A*, 2005). Furthermore, there are diurnal and nocturnal animals of which PRCs have a lower and higher A/D ratio, respectively (e.g. *Gallus* (diurnal) and *Mesocricetus auratus* (nocturnal); Johnson, *Chronobiol Int*, 1999).

However, it has been shown that the PRC of diurnal animals *generally* has a higher A/D ratio than that of nocturnal animals (Pittendrigh & Daan, *J Comp Physiol A*, 1976; Johnson *Chronobiol Int*, 1999). In particular, the previously reported PRCs of diurnal primates including humans show a higher A/D ratio than that of nocturnal mice, which are the most commonly used preclinical models (Hoban & Sulzman, *Am J Physiol*, 1985; Glass et al, *Am J Physiol*, 2001; Comas et al, *J Biol Rhythms*, 2006; Ruger et al, *J Physiol*, 2013; Hilaire et al, *J Physiol*, 2012). That's why we emphasized the difference of active time between mice, NHPs and humans in our manuscript. However, this can be misinterpreted as that the nocturnal and diurnal animals necessarily have different photosensitivity as the reviewer pointed out. To clearly describe these points, we have narrowed down our scope to nocturnal mice, diurnal NHPs and humans rather than all nocturnal and diurnal animals as follows:

- P7 L126: "Our work reveals a previously unrecognized biological variable in translating the efficacy of clock-modulating drugs from *nocturnal mice* to *diurnal humans*: their different photosensitivity."

- P18 L365: “Such a strong attenuating effect of light on a drug-induced phase delay is expected to exist in humans as their light PRC has a large magnitude of *the* advance zone (Ruger et al, 2013; St Hilaire et al, 2012). Thus, *the* phase delay induced by melatonin would also be expected to be strongly attenuated by light. This would explain its weak effectiveness in treating ASPD and jetlag after westward travel in a daylight cycle (Sack et al, 2007; Spiegelhalder et al, 2017; Zhu & Zee, 2012). These results indicate that such interspecies difference in photosensitivity should be considered *when translating the efficacy of clock modulating drugs from nocturnal mice to diurnal humans.*”

4) I am not entirely convinced by the authors' conclusion stating that "we found that such variability in the CK1i effect is mainly due to altered PER2 abundances" (page 16), because the analysis shown in figure 4 is limited to the correlation only between phase shift and PER2 level. By using the model, it would be reasonable to ask whether the PER2 level is the only factor that satisfactorily explains the different phase shift (by calculating the correlation between phase shift and many other parameters). Conversely, it would be possible to tune the parameters to change the expression level of PER2 but not change the free-running period and confirm the correlation against phase shift upon CK1i challenge.

We appreciate the reviewer for pointing this out. Indeed, previously, we have only shown the strong positive correlation between PER2 abundance and the effect of the CK1 δ/ϵ inhibitor (CK1i) (Fig 4C). To support that PER2 abundance is indeed a key determinant for the effect of CK1i, we have performed additional simulations suggested by the reviewer and added them to the manuscript.

First, while the correlation between PER2 abundance of the various ASPD models and the effect of CK1i have been investigated previously, this has now been extended for other clock proteins (Fig EV5B). We found that the abundance of PER2 is significantly more strongly correlated with the effect of CK1i than that of the other clock proteins (Fig EV5C). Next, we also found that the strong correlation between PER2 and the effect of CK1i is not due to the altered free-running period because the free-running period of the ASPD models and the effect of CK1i are weakly correlated (Fig EV5D). Furthermore, when we tuned a model parameter, which increases PER2 level in the model, regardless of its effect on the free-running period (Fig EV5E), the effect of CK1i becomes stronger (Fig EV5F). These additional results have now been included as follows:

- P15 L311: “To identify the source of the heterogeneous drug response among the ASPD models, we estimated the relationship between the effect of PF-670 (red squares; Fig 4B) and the average level of various core clock molecules of the ASPD models (Fig EV5B). Interestingly, *we found that the average protein level of PER2 is significantly more strongly correlated with the effect of PF-670 than that of the other clock proteins (Fig EV5C). Specifically, as the average PER2 protein levels (Fig 4A inset) increase in the ASPD models, the effect of PF-670 (red squares; Fig 4B) becomes stronger (Fig 4C). This correlation is not due to the different free-running period of the ASPD models (Fig EV5D) and appears to stem from the fact that phosphorylation of PER2 by CK1 δ/ϵ is the target of PF-670. Indeed, when we increase PER2 level in the model by tuning model parameters, the effect of CK1i becomes*

stronger regardless of its effect on the free-running period (Fig EV5E and F).”

Figure EV5. The response of the ASPD models to light and PF-670: PER2 level is strongly correlated with the effect of CK1i.

B The correlation between the effect of PF-670 (red squares; Fig 4B) and the average level of various core clock proteins of the ASPD models (Fig 4A inset and Dataset EV3). The line represents the least-square fitting line. r and α denote the Pearson's correlation coefficient and the slope of the least-square fitting line.

C The average protein level of PER2 is significantly more strongly correlated with the effect of PF-670 than that of other clock proteins. * and ** indicates $P < 0.05$ and $P < 0.001$, respectively. Here, P-value is estimated by Pearson's correlation test.

D The correlation between the effect of PF-670 (red squares; Fig 4B) and the free-running periods of the ASPD models (Fig 4A) is weak and not significant.

E, F As PER2 level increases in the model, regardless of the change in the free-running period (E), the effect of CK1i becomes stronger (F). The mRNAⁿ (orange line), Per2^t (green line) and Cry1^d (blue line) were simulated by perturbing the parameters tmc , $trPt$ and uro , which describe the nuclear export rate of mRNA, the transcription rate for Per2, and the degradation rate for CRY1, respectively (see Dataset EV1 for details). The line and colored range in F represent the mean \pm SEM of the phase delays induced by a single 20 mpk DD dosing at CT1, 2, 3, ..., 24.

G The ratio of average PER2 levels in LD 8:16 and LD 16:8 when dosing occurs (red squares; Fig 4B) is higher than 1 in all ASPD models.

It would also be important to show that the current model recapitulates the correlation shown in figure 4E.

We also performed additional simulations to test whether the correlation shown in Fig 4E is captured by our model (Fig 2A). Indeed, consistent with the experimental data, the model also simulates that the effect of CK1i becomes stronger as PER2 abundance becomes higher at the dosing time. Note that the experimental data and simulations do not completely match as the experiments were done in rat and simulations were performed with the NHP model:

- P15 L321: “Furthermore, PER2 abundance also explains the different effect of PF-670 depending on day length: due to the higher PER2 abundance at the dosing times in LD 8:16 than in LD 16:8 (Fig EV5G), a larger PF-670-induced phase delay is simulated in LD 8:16 than in LD 16:8 (Fig 4D). To support these *in silico* predictions (Fig 4C and D), we estimated the relationship between the effect of PF-670 and PER2 levels from experimentally measured PRC to PF-670 (Badura et al, 2007) and time series of PER2 levels in the SCN (Amir et al, 2004). Indeed, we found a strong positive correlation between them, *which is also recapitulated by our model (Fig 4E).*”

Figure 4. CK1i induces a larger phase delay as PER2 levels increase depending on the molecular cause of ASPD and external lighting conditions.

E Consistently, higher experimentally measured PER2 levels in the SCN (Amir et al, 2004) at the dosing time leads to a larger PF-670-induced phase delay (Badura et al, 2007), *which was also captured by the NHP model when 20 mpk dosing was used.*

Related to this, please explain the reason why the phase shift did not change between ZT4 dosing and ZT11 dosing (figure 2) where the level of PER2 should change in wildtype NHPs.

We thank the reviewer for pointing this out. Indeed, as PER2 level is expected to be lower at CT4 than CT11 in NHPs, the effect of DD dosing at CT4 is weaker than that at CT11 (4.4 h; CT4 and 5.9 h; CT11). However, the effects of CK1i dosing at ZT4 and ZT11 under LD become similar in NHPs (1.5 h; ZT4 and 1.7 h; ZT11) due to the strong attenuating effect of light. Specifically, the range of drug effect depending on dosing time (from 0.8 and 6.1hr in DD) is dramatically reduced by light (from 0.6 and 2.1hr in LD). To describe this, we have added Fig EV3 (see our response to comment 2 above) and text in the manuscript as follows:

- P11 L226: “We next investigated whether the model can *predict* the combined effect of PF-670 and light even when the dosing time changes. We chose dosing at ZT4 as it leads to a much weaker phase delay in mice than dosing at ZT11 (Badura et al, 2007; Kim et al, 2013). However, in NHPs, dosing at ZT4 led to a similar phase delay as dosing at ZT11 (1.5; ZT4 and 1.7 h; ZT11; Fig 2B and F). This unexpected drug effect in NHPs was accurately predicted by the model with the new light module (Fig 2F), *which is mainly due to the strong attenuation of the drug effect by light in NHPs (Fig EV3).* Taken together, the difference in light response between mice and NHPs is a critical factor leading to their heterogeneous response to a clock-

modulating drug.”

Minor comments:

5) Abstract: "a systems pharmacology model" does not summarize any model structures. Please briefly mention what types of model (e.g., a model describing the detailed molecular reactions, etc.) was used in this study. Note that I am not requesting to remove the word "systems pharmacology model."

We completely agree with the reviewer that “a systems pharmacology model” does not describe our model (Fig 2A) in detail enough. Thus, we have added its description in the abstract as follows:

- P3 L41: “a systems pharmacology model *describing molecular interactions*”

Furthermore, as we were not able to add a further detailed description in the abstract due to its strict length limit (175 words), we have also added the model description in the introduction:

- P6 L120: “The model simulations *for the intracellular interactions of PF-670 with clock components*”

6) Figure 4C and 4D: please indicate the points of each mutation (not just by showing the name of mutants). Also, several overwrapped characters (name of CRY mutants) should be amended.

We thank the reviewer for the comment. We have now revised the Figure 4C and D as suggested to improve the clarity of the figure.

Figure 4C and D

7) The adaptive chronotherapeutic approach shown in figure 5 works well. However, one may naively think that the iterative adaptation works in any case without the help of model prediction. Please consider providing more rationale and discussion about how the model prediction provides the basis of the adaptive chronotherapeutic protocol introduced in this

study.

We agree with the reviewer and have added the following rationale about why the model simulation is useful to test whether the adaptive chronotherapeutic strategy works as expected (Fig 5A and EV6A):

- P16 L336: “By using this feature, we developed an adaptive chronotherapeutic approach: if the current dosing regimen leads to a weaker or stronger drug effect than the desired one, the dosing time is delayed or advanced by 1 h, respectively until the desired phase delay is achieved (Fig 5A and EV6A).

To test whether this approach works as expected despite the large perturbation of the circadian clock (e.g. PER2 abundance and phase) by genetic variation or environmental lighting conditions, we applied the adaptive chronotherapeutic approach to all ASPD models with varying day lengths. Specifically, a single daily 10 mpk dose was given at ZT3 to the ASPD models in LD 16:8 (Day 1; Fig 5B). Then, depending on the induced phase delay, the initial dosing time (ZT3) was adjusted according to the chronotherapeutics (Fig 5A and EV6A).”

8) This is a very minor comment for figure 5A; what do the sun and snow marks at the upper-left of humans represent?

We used the marks of the sun and snow to describe long and short days, respectively (i.e. summer and winter). But, as the reviewer pointed out, the marks do not clearly represent the intended meaning, thus we have changed the sun and snow marks to the following markers representing short and long days.

Figure 5A

Reviewer #3:

The authors address an interesting and relevant interdisciplinary topic - the treatment of advanced sleep phase disorder (ASPD) via CKI inhibitor. The authors compare novel primate data with previously published mouse data (Fig.1) and adapt existing mathematical models accordingly (Fig.2). It is suggested that differential gating mechanisms modulate the effects of CKI inhibitors. Consequently, treatment of ASPD cannot be based only on mouse experiments. In addition, gating models and some data indicate that PER2 levels influence the phase shifts

via inhibitor (Fig.4). The systemic understanding of treatment allows predictions of relatively simple strategies to adapt dosing regimen (Fig.5).

My major concerns refer to the limits of quantitative modeling. In my eyes, there is not a single quantitative model of a eukaryotic clock since multiple transcription factors (see Ueda reviews), epigenetic regulations (Sassone-Corsi, Takahashi), huge protein complexes of unknown stoichiometry (Weitz), multiple phosphorylations (Virshup, Kramer) etc. do not allow detailed and precise modeling. The last author, however, is well aware of the limitations of modeling and employs ensembles of reasonable models (Methods, Appendix). Consequently, the main results (phase shifts, differences between night- and day-active animals, role of PER2) seem to be independent of modeling details. Thus the proposed treatment strategy seems quite useful despite known limitations of quantitative modeling.

We are grateful to the reviewer for the positive feedback on our work despite the limitation of the modeling in general. Indeed, we agree with the reviewer about the complexity in the circadian clock. We have now added the suggested references describing such complexity.

Specific comments:

1. The known phase shift due to electrical illumination should not be called shift work.

We thank the reviewer for pointing this out. We have revised the manuscript to only focus on emphasizing the high prevalence of circadian disruption by referring to epidemiological studies on shift workers as follows:

- P4 L71: “The failure of synchrony between the clock and external cycles can occur due to dysfunction of the circadian clock system or alteration of the external environment. *Notably, recent epidemiological data suggest that more than 80% of the population appears to live a shift work lifestyle (Sulli et al, 2018)*”.

2. Page 7 "... these processes are slowed down" The connection of to PER2 stability, nuclear import and export and periods are quite complex (see, e.g., papers by Vanselow and Relogio). Period shortening and lengthening can be explained by similar mechanisms.

We agree with the reviewer on this. For instance, the *in-silico* study showed that the increase in the degradation rate of Per mRNA could lead to both shorter and longer period (Relogio et al, PLoS Comput Biol, 2011). Furthermore, the experimental study showed that decreased nuclear import of PER alters the period differently: depletion of Importin β and Transportin 1, which decrease nuclear import of PER, lengthens and shortens period, respectively (Korge et al, PLoS Genet, 2018). Thus, it is risky to conclude that the slowing-down of PER1/2 degradation, their binding to CRY1/2, and nuclear translocation *always* lead to the period lengthening and delaying the phase as the reviewer pointed out. Thus, we have revised the text and Fig 1A in a way summarizing the previously reported effect of PF-670 on the circadian clock as follows:

- P8 L142: “PER1/2 phosphorylation by CK1 δ/ϵ regulates their degradation, binding to CRY1/2, and nuclear translocation, which are the key processes of the transcriptional-translational negative feedback loop of the mammalian circadian clock (Fig 1Ai) (Ode & Ueda, 2018; Gallego & Virshup, 2007). When CK1 δ/ϵ is inhibited by PF-670, these processes are slowed down (Fig 1Ai) and circadian phase is delayed, which is attenuated by light, the strongest zeitgeber (Fig 1Aii) (Badura et al, 2007; Kim et al, 2013; Meng et al, 2010; Sprouse et al, 2010; Walton et al, 2009).”

Figure 1. Light attenuates the effect of PF-670 more strongly in diurnal NHPs than in nocturnal mice.

A PF-670 inhibits the phosphorylation of PER by CK1 δ/ϵ (i) and delays the circadian phase, which is counterbalanced by light (ii). Thus, daily dosing leads to continually accumulating phase delay in DD and constant stable phase delay in LD.

3. Recently, expression profiles in baboon data have been published providing despite limited sampling some reasonable phases and amplitudes of core clock genes. Are the differences to mouse data (e.g. from Hogenesch) connected to the discussed differential phases responses?

As the reviewer mentioned, Mure and his colleagues have recently published the peak time of clock genes expression in the SCN of baboons and compared this with that of mice (Mure et al, Science, 2018). Unfortunately, Mure et al. did not report the peak time of Per2, but they found that the peak time of Per1 mRNA expression is ~5 h more advanced in the SCN of baboons than in that of mice under LD. Thus, the peak time of Per2 is also expected to be advanced in baboons than mice. Consistently, our NHP model simulates a ~2 h more advanced phase of Per1 and Per2 mRNA expression than the mouse model (Kim et al, CPT:PSP, 2013) (Fig R2A). Reflecting the phase difference in clock gene expression, the simulated PRC to the CK1 δ/ϵ inhibitor by the NHP model is more advanced than that by the mouse model (Fig R2B).

Figure R2. As the phase of clock gene expression in the NHP model is more advanced compared to that in the mouse model (Kim et al, CPT:PSP, 2013), the PRC to CK1 δ/ϵ inhibitor is more advanced in the NHP model than in the mouse model. **A** The phase of Per1 and Per2 mRNA abundance simulated by the NHP model (Fig 2A) is more advanced than that simulated by the mouse model (Kim et al, CPT:PSP, 2013) under LD 12:12 due to the different photosensitivity. **B** Due to the advanced phase of clock gene expression in the NHP model than in the mouse model, the PRC to CK1 δ/ϵ inhibitor is more advanced in the NHP model than in the mouse model. The PRC to the 3-day LD 10 mpk dosing and the 3-day LD 32 mpk dosing were simulated by the NHP model (red line) and the mouse model (blue line), respectively.

Although the interspecies difference in the phase of clock gene expression can cause the interspecies variations in the effect of the CK1 δ/ϵ inhibitor (Fig. R2), this is unlikely the major reason for the interspecies difference in the attenuating effect of light we observed (Fig. 1E-G and 3A). Specifically, by reflecting the more advanced phase of clock gene expression in the SCN of baboons than in that of mice (Mure et al, Science, 2018), we compared the effect of the CK1 δ/ϵ inhibitor at different times between NHPs and mice (i.e. more advanced time in NHPs than mice): CT4; DD and ZT4; LD for NHPs and CT11; DD and ZT11; DD for mice. Despite the more advanced dosing timing for NHPs, the attenuation of the drug effect by light is still stronger in NHPs (2.9 h; red arrow; Fig R3) than in mice (1.7 h; blue arrow; Fig R3). Thus, it is hard to conclude that the interspecies difference in the counteracting effect of light is mainly due to the different phase of clock gene expression between NHPs and mice.

Figure R3. Although the dosing time is more advanced in NHPs than in mice, reflecting the more advanced phase of clock gene expression in the SCN of baboons than in that of mice (Mure et al, Science, 2018), the attenuation of the drug effect in LD is stronger in NHPs (2.9 h) than in mice (1.7 h). For DD and LD dosing, NHPs were dosed with 10 mpk PF-670 at CT4 and ZT4 for 3 days, respectively (Fig 1E and 2F and EV1C and EV3), and mice were dosed with 32 mpk PF-670 at CT11 and ZT11 for 3 days, respectively (Fig 1E and F) (Kim et al, CPT:PSP, 2013).

Although the stronger attenuation of the drug effect by light in NHPs than in mice does not appear to be mainly caused by their different phase of clock gene expression, we did feel that the comment that the phase of clock gene expression needs to be also considered as a factor leading to the interspecies variability in drug effect is important. So, we have now discussed the effect of the different phase of clock gene expression on the phase delay induced by the CK1 δ/ϵ inhibitor as follows:

- P18 L380: “We found that such variability in the CK1i effect is mainly due to altered PER2 abundances (Fig 4C-E and EV5B-F). *While the interspecies difference in PER2 abundance has not been investigated, it may contribute to the interspecies variability in the CK1i effect (Fig 1E-G and 3A). Furthermore, the interspecies difference in the phase of PER2 rhythms (Millius & Ueda, 2018; Mure et al, 2018; Vosko et al, 2009; Zhang et al, 2014), which leads to the interspecies difference in PER2 abundance when dosing occurs, could also be the source of the interspecies variability in the CK1i effect. It would be interesting to investigate such a relationship between the effect of CK1i and PER2 abundance, and extend this to other clock-modulating drugs and their target molecules (e.g. KL001 targeting CRY (Hirota et al, 2012)).*”

Furthermore, we have described the more advanced phase of clock gene expression in the NHP model (Fig 2A) than in the mouse model (Kim et al, CPT:PSP, 2013) under LD in the manuscript:

- P29 L594:

“Model assumptions

(1) The model of the intracellular mammalian circadian clock, Kim-Forger model, which was developed to accurately simulate the mouse SCN (Kim & Forger, 2012), was used for the new NHP model as clock gene expression profiles in the SCN of cynomolgus monkeys have not been measured. Nevertheless, due to the new light module, the NHP model simulates more advanced clock gene expression compared to the original mouse model under LD, which is consistent with the advanced clock gene expression seen in the SCN of baboons compared to that of mice under LD (Millius & Ueda, 2018; Mure et al, 2018).”

4. Some comments on the number of new model parameters, fitting procedures, limitations of models might be added to the main text. Without reading previous papers of the last author and of the Appendices it is difficult to develop an understanding of the underlying models (e.g. the meaning of models N and W are unclear from the main text).

We agree with the reviewer that the detailed model descriptions, such as the number of new model parameters, fitting procedure and limitations of the model (i.e. underlying model assumptions), need to be included in the main text as it helps readers more easily understand the model. However, the current manuscript is already quite long, as pointed out by reviewer 1, it is difficult to include the detailed description in the main text. Thus, we have decided to

provide the summarized description of the model in the main text with clear references for the location of detailed descriptions. Furthermore, we have added Appendix Equation S1, Table EV1 and 2 and Dataset EV1-3 to describe the model equations, variables and parameters, respectively. We have also revised the Materials and Methods to describe the model more clearly and to include the underlying model assumptions.

- P9 L176: “NHPs and mice show large differences in *the* pharmacokinetics of PF-670 (Fig 1B), and the effect of PF-670 on circadian phase, and how this is influenced by light (Fig 1E-G). To analyze such multiple differences systematically, we developed the first systems chronopharmacology model for NHPs by modifying our previous model (Kim et al, 2013), which successfully simulates the effects of PF-670 and light on the intracellular circadian clock of mice. *The modified parts of the model including the newly estimated pharmacokinetic parameters and new equations for the light module are described in the Materials and Methods. See Appendix Equation S1, Table EV1 and 2 and Dataset EV1 and 2 for the detailed description for the equations, variables, and parameters of the NHP model.* In the NHP model, inhibition of CK1 δ/ϵ for PER1/2 phosphorylation by PF-670 (Fig 2Ai) and light-induced Per1/2 gene transcription via CREB (Fig 2Aii) are incorporated to simulate the resulting phase shift of the circadian clock at the molecular level (orange arrow; Fig 2Aiii).”

- P26 L539:

“**Modeling Studies**

Mathematical model description

We adopted our previous systems chronopharmacology model (*274 variables and 86 parameters*), which successfully investigated the effect of PF-670462 in mice (Kim et al, 2013). This model was developed by extending the Kim-Forger model (Kim & Forger, 2012), a detailed mathematical model of the intracellular mammalian circadian clock, *which* describes the reactions among core clock molecules (e.g. binding, phosphorylation, subcellular translocation, transcription, and translation) using ordinary differential equations based on mass action kinetics (181 variables and 75 parameters). *To develop the systems chronopharmacology model for NHP (Fig 2A), the original mouse model (Kim et al, 2013) was modified by newly estimating the pharmacokinetic (PK) parameters (Fig 2Ai) and incorporating the gating and adaptation into the light module of the model (Fig 2Aii).*

Modification of the PK parameters

The six parameters describing the PK properties of PF-670462 (e.g. transfer rate between plasma and brain tissue) were modified due to the difference in free PF-670462 exposure in brain tissues (Fig 1B) and its effect in DD (Fig 1D) between NHPs and mice. Specifically, the parameters were fitted to the disposition profiles of PF-670 and its DD dosing effect in NHPs (see Fig EV1A-C and Dataset EV1 for details).

Incorporation of gating and adaptation for light into the model

To incorporate the gating into the model, we used the function g , which determines the photosensitivity of the circadian clock at each CT (Fig EV1F). To connect the photo-insensitive and photosensitive zones of g , a piecewise polynomial interpolation was used (see code EV1).

To incorporate the adaptation into the model, we used the Hill function, which expresses light duration-dependent reduction of photosensitivity (Fig EV1H).

The values of six parameters determining the functions of gating and adaptation (Dataset EV2) were estimated with the simulated annealing (SA) method (Gonzalez et al, 2007) and post filtering in two steps. In the first round, we found 991 parameter sets with which the model accurately simulates the phase delay of NHPs induced by the 3-day LD dosing and the magnitude of human PRC to a 6.7 h light pulse (Fig EV2A) using the SA method with the cost function:

$$2\sqrt{\left(\frac{\min(m-\tilde{m}+1,0)}{\tilde{m}-1}\right)^2 + \left(\frac{\max(m-\tilde{m}-1,0)}{\tilde{m}+1}\right)^2 + \left(\frac{\min(M-\tilde{M}+1,0)}{\tilde{M}-1}\right)^2 + \left(\frac{\max(M-\tilde{M}-1,0)}{\tilde{M}+1}\right)^2} + f_1 + \sqrt{\sum_{j=1}^3 \left(\frac{x_j - \tilde{x}_j}{\tilde{x}_j}\right)^2}$$

$$\text{where } f_1(x) = \begin{cases} 10^3 & \text{If } \max(|m|, |M|) \geq 6 \\ 0 & \text{If } \max(|m|, |M|) < 6 \end{cases}$$

m and M are the min and max of the simulated PRC to a 6.7 h light pulse, respectively. $\tilde{m} = -3.26$ and $\tilde{M} = 2.63$ h are the min and max of the human PRC to a 6.7 h light pulse, respectively (Khalsa et al, 2003). x_j and \tilde{x}_j are the simulated phase delay and experimentally measured phase delay (Fig 1D) on day j of 3-day LD dosing, respectively.

In the second round, using post filtering, among the 911 parameter sets estimated by the SA method, we identified the 10 parameter sets (Dataset EV2) with which the model accurately simulates the type 1 PRC to a 12 h light pulse and the human PRC to a 6.7 h light pulse and to 3-cycle 5 h light pulses (Fig EV2).

To estimate the input CT for g even when the circadian phase is perturbed, we constructed the function p which determines the CT from an internal pace marker: the phase angle of the limit cycle of two clock variables, $revng$ and $revnp$ in the model (Fig EV1G). We first interpolated the phase angles of the limit cycle of $revng$ and $revnp$ to the CTs. Then, we composed the interpolation function with the function, $\tan^{-1}(revng(t), revnp(t))$, which estimates the phase angle of the limit cycle from the concentrations of $revng$ and $revnp$ at t . As $g \cdot p$ is the composite of the interpolation functions, which do not have explicit form, including $g \cdot p$ into the model increases the computational cost of the simulation. Thus, the approximated $g \cdot p$ using Fourier series was used, which accurately determines the phase-dependent photosensitivity even when the circadian phase is altered by PF-670462 or light.

Development of the ASPD models

To develop the ASPD models (Dataset EV3), we investigated that the modification of which parameter allows for advancing the phase by ~4 h from the WT model (Fig 4A) reflecting the advanced circadian phase of ASPD patients (Jones et al, 1999). In this process, to reflect the advanced circadian phase, the phase of gating function g is also advanced by 4 h.

Model assumptions

(1) The model of the intracellular mammalian circadian clock, Kim-Forger model, which was developed to accurately simulate the mouse SCN (Kim & Forger, 2012), was used for the new NHP model as clock gene expression profiles in the SCN of cynomolgus monkeys have not been measured. Nevertheless, due to the new light module, the NHP model simulates more advanced

clock gene expression compared to the original mouse model under LD, which is consistent with the advanced clock gene expression seen in the SCN of baboons compared to that of mice under LD (Millius & Ueda, 2018; Mure et al, 2018).

(2) The pharmacodynamic parameters, describing the intracellular action of PF-670462 (e.g. binding of PF670 with CK1 δ/ϵ), of the original mouse model (Kim et al, 2013) were kept as they are expected to be similar between NHPs and mice.

(3) The reduced photosensitivity due to adaptation for light during daytime is assumed to be completely recovered after nighttime if it is long enough (≥ 8 h).

(4) To simulate the phase shift of activity onsets with the model, we used the phase shift of the simulated BMAL1 gene expression profile as the phase of BMAL1 gene expression profiles and activity onset are highly correlated (Kiehl et al, 2010). However, as the phases of all clock gene expression profiles are tightly interlocked in the model, the result changes little even if the phase of other clock gene expression profiles is used.”

Furthermore, we have revised the text and Fig 3E to clearly describe the meaning of models N and W as follows:

• P12 L257: “Given the strong attenuating effect of light on CK1 δ/ϵ inhibition in humans (Fig 3C), we would expect a potentially large variation in CK1i response in individuals with different levels of photosensitivity (Fig 3D). To investigate this, among the 10 pairs of gating and adaptation (Fig 2Aii), two pairs were chosen; one has a narrower high photosensitivity zone than the other (Fig 3E inset) (see Fig EV4A and Dataset EV2 for details). Thus, the model with the narrow high photosensitivity zone (model N; Fig 3E) simulates the smaller magnitude of the advance zone of light PRC than the model with the wide high photosensitivity zone (model W; Figure 3E).”

Figure 3. Inter- and intraspecies variability in photosensitivity causes variation in the effect of PF-670.

E Model N with a narrow high photosensitivity zone (inset) simulates a smaller magnitude of

the advance zone of 12 h light PRC than model W with a wide high photosensitivity zone.

5. It might be stressed more clearly how data have been used to constrain models (training sets) and what data are consistent with experiments without explicit fitting.

We agree with the reviewer that the clear explanation of how training sets have been used to constrain the model and what experimental data are consistent with the model simulation without fitting (test sets) is important for readers to easily check the reliability of the model. So, we have revised the manuscript as follows:

- P11 L216: “As the new light module of the model was estimated mainly based on the response of humans to light (*Fig EV2*), we next investigated whether it could accurately *predict* the potent light-induced attenuation of *the* PF-670 effect in NHPs.”
- P11 L221: “We found that such light-induced phase shifts *that occur* during and after jet lag were accurately *predicted* by the model with the new light module (*Fig 2D*).”
- P11 L226: “We next investigated whether the model can *predict* the combined effect of PF-670 and light even when the dosing time changes.”
- P27 L554: “*Specifically, the parameters were fitted to the disposition profiles of PF-670 and its DD dosing effect in NHPs (see Fig EV1A-C and Dataset EV1 for details).*”
- P51 L1138: “The model accurately predicts the human PRC to a 3 h light pulse adopted from (Minors et al, 1991), *which is not used in the estimation process (H).*”

6. It should be discussed how the acute circadian phase can be estimated to adapt control. There are traditional techniques (activity, melatonin onset, body temperature (R. Wever book)) and newly developed markers (Ueda, Dallmann, Kramer).

We totally agree with the reviewer that the accurate estimation of circadian phase should be discussed as it is required to use our adaptive chronotherapeutics (*Fig 5A and EV6A and B*). Thus, we have added the following sentence in the discussion to introduce techniques for the accurate estimation, which have been recently developed by the Hiroki Ueda group, Achim Kramer group and Daniel Forger group (Kasukawa et al, PNAS, 2012; Walch et al, Sci Adv, 2016; Wittenbrink et al, JCI, 2018).

- P19 L396: “However, because obtaining such information can be challenging, we developed an adaptive chronotherapeutics, which identifies the precise dosing time to achieve normal circadian phase by tracking the patient’s drug response (*Fig 5A and EV6A and B*). *This adaptive chronotherapeutics requires a precise quantification of the drug-induced circadian phase shift in daily life. Recently, to overcome the insufficient accuracy of actigraphy and labor-intensive measurement of dim light melatonin onset, various methods have been developed, which are based on the metabolite timetable (Kasukawa et al, 2012), one-time phase estimation (Wittenbrink et al, 2018), or sleep-tracking using mobile phone (Walch et al, 2016). With these advances, the adaptive chronotherapeutics could play a critical role in incorporating biomarker data and providing real-time patient-tailored chronotherapy via smart devices (i.e. digital medicine) (Elenko et al, 2015).*”

Thank you for sending us your revised manuscript. We have now heard back from the three reviewers who were asked to evaluate your study. As you will see the reviewers are now overall supportive and I am pleased to inform you that your manuscript will be accepted in principle pending the following essential amendments:

- To enhance reproducibility and add value to papers including mathematical models, we are offering a "model curation service" in collaboration with Prof. Jacky Snoep and the FAIRDOM team. In the process of verifying the model of your manuscript, Jacky encountered difficulty in reproducing some of your modeling results. We would therefore kindly ask you to carefully consider the points noted in the technical report below (*Model Curation Report*) and to fix these issues when you submit your revision.

- Once the code has been fixed, please deposit your computational code and primary datasets to an appropriate public database and provide a resolvable link to the dataset in the Data Availability section accordingly.

****Model Curation Report**:**

For the model description, the authors focus on the extensions that were made to the original model, which is published in another paper. This is understandable, but makes it quite hard to understand the model, when one has not studied the previous manuscript.

The model is made available as a Mathematica notebook, and the authors provide a word document with hints for reproduction of some of the figures. Due to the size of the model it is still difficult to know how to make the necessary changes to simulate the different figures in the manuscript. I would like to request the authors to make small changes to the Mathematica notebook, to specify what value to set for the "select" variable, to simulate the different figures. The select value should be the only thing that the user should change, i.e. it should then also ensure the correct settings for "days", "dtt", "dose" and "ldd". There should be a key stating, chose value x for "select" to simulate figure y in the manuscript.

When I simulated the "PF-670462_NHP_model" notebook as is, I obtained a final figure that resembles "model N" in Figure 3F, which seems to be in agreement with the comments in the notebook. I am therefore quite confident that it is possible to simulate the other figures as well, my only request is that the authors make it easier to do this, as described above.

When I simulated the "ASPD_models.nb" I obtained results with a positive phase shift, which looked quite different from the original results. I analysed the notebook in Mathematica 12.0 which is a newer version than the authors used, which could possibly have resulted in a different result, but I did not see any error messages, or warnings. As for the other notebook, it would be nice if the authors could make the selection of simulation of the different mutants, as easy as possible, i.e. indicate what value of "select" to chose to simulate a specific figure (no additional setting of other parameters).

REFEREE REPORTS

Reviewer #1:

The authors have revised the manuscript to address the issues raised in the previous review round and as such the paper is now suitable for publication.

Reviewer #2:

The authors did an excellent job to answer all of my comments adequately. This manuscript is ready for publication. Just a very minor point: line 612: the CPU clock might be 1.50 GHz x 8, not 150 GHz x 8.

Reviewer #3:

The authors addressed carefully all comments of the reviewers. They found a good compromise to be more precise regarding modelling without expanding the manuscript too much.

- To enhance reproducibility and add value to papers including mathematical models, we are offering a "model curation service" in collaboration with Prof. Jacky Snoep and the FAIRDOM team. In the process of verifying the model of your manuscript, Jacky encountered difficulty in reproducing some of your modeling results. We would therefore kindly ask you to carefully consider the points noted in the technical report below (*Model Curation Report*) and to fix these issues when you submit your revision.

Thanks for pointing this out. We found that one line of the code was missed. We have now added the missed one to the code, which allows for successful reproduction of all the simulations in the manuscript. Furthermore, we have added a text annotation to the code for better readability. Please see our response to the model curation report below for details.

- Once the code has been fixed, please deposit your computational code and primary datasets to an appropriate public database and provide a resolvable link to the dataset in the Data Availability section accordingly.

We have now deposited the computational code on Github and the link has been provided in the Data availability section in the manuscript as follows:

- P3 628: “The MATHEMATICA codes used in this study are available in code EV1 and the following database: <https://github.com/daewookkim/Non-human-primate-circadian-clock-model-including-CKI-inhibitor>.”

****Model Curation Report**:**

For the model description, the authors focus on the extensions that were made to the original model, which is published in another paper. This is understandable, but makes it quite hard to understand the model, when one has not studied the previous manuscript.

The model is made available as a Mathematica notebook, and the authors provide a word document with hints for reproduction of some of the figures. Due to the size of the model it is still difficult to know how to make the necessary changes to simulate the different figures in the manuscript. I would like to request the authors to make small changes to the Mathematica notebook, to specify what value to set for the "select" variable, to simulate the different figures. The select value should be the only thing that the user should change, i.e. it should then also ensure the correct settings for "days", "dt", "dose" and "l". There should be a key stating, chose value x for "select" to simulate figure y in the manuscript.

When I simulated the "PF-670462_NHP_model" notebook as is, I obtained a final figure that resembles "model N" in Figure 3F, which seems to be in agreement with the comments in the notebook. I am therefore quite confident that it is possible to simulate the other figures as well, my only request is that the authors make it easier to do this, as described above.

We agree with Prof. Jacky Snoep and the FAIRDOM team that clear explanation of how the figures in the manuscript were simulated using the computer code is important. Thus, we have clearly described what setting of the input parameters (e.g. "dt" and "dose") allows to simulate the figures by adding a text annotation at the beginning of the notebook of "PF-670462_NHP_model" and "ASPD_models" (code EV1) as follows:

• "PF-670462_NHP_model.nb"

Guideline to simulate the PF-670-induced phase delays in Fig 2B, F, 3F and 4B

Fig 2B: Set modeltype=N, select=1~3, days=4, ldd=12, dose=10 and dt=11 or set modeltype=W, select=1~7, days=4, ldd=12, dose=10 and dt=11.

Fig 2F: Set modeltype=N, select=1~3, days=4, ldd=12, dose=10 and dt=4 or set modeltype=W, select=1~7, days=4, ldd=12, dose=10 and dt=4.

Fig 3F model N: Set modeltype=N, select=1, days=10, ldd=12, dose=25 and dt=8.

Fig 3F model W: Set modeltype=W, select=1, days=10, ldd=12, dose=25 and dt=8.

Fig 4B WT model:

Blue circles

1. Set modeltype=N, select=1, days=21, ldd=12, dose=26 and dt=4
2. Set modeltype=N, select=1, days=21, ldd=12, dose=24 and dt=5
3. Set modeltype=N, select=1, days=21, ldd=12, dose=22 and dt=6
4. Set modeltype=N, select=1, days=21, ldd=12, dose=19 and dt=7

- “ASPD_models.nb”

Guideline to simulate the PF-670-induced phase delays in Fig 4B

Fig 4B ASPD models:

Blue circles

1. Set select=1~8, days=21, ldd=12, dose=26 and dtt=4
2. Set select=1~8, days=21, ldd=12, dose=24 and dtt=5
3. Set select=1~8, days=21, ldd=12, dose=22 and dtt=6
4. Set select=1~8, days=21, ldd=12, dose=19 and dtt=7

Red circles

1. Set select=1~8, days=21, ldd=12, dose=26 and dtt=0
2. Set select=1~8, days=21, ldd=12, dose=24 and dtt=1
3. Set select=1~8, days=21, ldd=12, dose=22 and dtt=2
4. Set select=1~8, days=21, ldd=12, dose=19 and dtt=3

Furthermore, we have located all of the input parameters at the beginning of the code for readers to easily use the model as follows:

- “PF-670462_NHP_model.nb”

Setting of model and dosing regimen to simulate PF – 670 – induced phase shift

```
(*Type of model: N → model N and W → model W*)
modeltype = N;

(*Select the parameter sets of gating and adapataion for light;
Each parameter set consist of 6 parameter {X1, X2, Y1, Y2, Z1, Z2};
Detailed description of the each parameter can be found in Table EV4;
Model N has 3 parameter sets of gating and adaptation for light;
Thus, please choose value of select between 1 and 3;
Model W has 7 parameter sets of gating and adaptation for light;
Thus, please choose value of select between 1 and 7;*)
select = 1;

(*Dosing Days*)
days = 10;

(*Light Duration (h)*)
ldd = 12;

(*dose level (mpk)*)
dose = 25;

(*Dosing Time (h)*)
dtt = 8;
```

- “ASPD_models.nb”

Setting of dosing regimen to simulate PF – 670 – induced phase shift

```
(* Selection of mutation, which induces advanced sleep phase disorder (ASPD)
1 → Per2t, 2 → Per2p, 3 → Per2n, 4 → Gsk3β3, 5 → Cry2t, 6 → Cry1t, 7 → Cry1d and 8 → B:Cp*)
select = 1;

(*Dosing Days*)
days = 21;

(*Light Duration (h)*)
ldd = 12;

(*dose level (mpk)*)
dose = 26;

(*Dosing Time (h)*)
dtt = 4;
```

When I simulated the "ASPD_models.nb" I obtained results with a positive phase shift, which looked quite different from the original results. I analysed the notebook in Mathematica 12.0 which is a newer version than the authors used, which could possibly have resulted in a different result, but I did not see any error messages, or warnings. As for the other notebook, it would be nice if the authors could make the selection of simulation of the different mutants, as easy as possible, i.e. indicate what value of "select" to chose to simulate a specific figure (no additional setting of other parameters).

Thanks for pointing this out. We found that we missed a line of code in "ASPD_models.nb", which calculates the phase shift. Thus, the calculated phase shift becomes positive although the simulated phase of gene expression profile is indeed delayed by PF-670 dosing (Figure R1). We have added the missed line and thus the revised code correctly calculates the phase shift which is consistent with Fig 4B in the manuscript (Figure R2). Furthermore, we have added the text annotation to the notebook of "ASPD models" to indicate what setting of the input parameters allows the model to simulate figures in the manuscript as mentioned above.

Figure R1. The ASPD model simulates the delayed phase of gene expression profile when CK1δ/ε inhibitor is treated. Here, the phase shift induced by 26 mpk daily single dosing at ZT4 in LD

12:12 was simulated using the ASPD model with Per2^t mutation (see Dataset EV3 for the description of Per2^t mutation).

Figure R2. Revised code correctly calculates the phase shift induced by CK1δ/ε inhibitor, which is consistent with Fig 4B.

Reviewer #1:

The authors have revised the manuscript to address the issues raised in the previous review round and as such the paper is now suitable for publication.

Reviewer #2:

The authors did an excellent job to answer all of my comments adequately. This manuscript is ready for publication. Just a very minor point: line 612: the CPU clock might be 1.50 GHz x 8, not 150 GHz x 8.

We thank the reviewer for pointing this out. We have now revised the description of the CPU clock as follows:

- P30 L615: “All the simulations and parameter searches were *performed* using MATHEMATICA 11.0 (Wolfram Research, Champaign, IL) *with a computer cluster composed of seven machines where each machine is equipped with two Intel Xeon SP-6148 CPUs (2.4GHz, 20C), 192GB RAM, and the operating system CentOS 7.4 64bit.*”

Reviewer #3:

The authors addressed carefully all comments of the reviewers. They found a good compromise to be more precise regarding modelling without expanding the manuscript too much.

Accepted

3rd June 2019

Thank you again for sending us your revised manuscript. We are now satisfied with the modifications made and I am pleased to inform you that your paper has been accepted for publication.

Corresponding Author Name: Jae Kyoung Kim
 Journal Submitted to: Molecular Systems Biology
 Manuscript Number: MSB-19-8838